# The Argmax Gap in Human Chess Move Prediction

## Abstract

Human-move predictors are typically evaluated by whether their highest-ranked move matches the move played. This evaluation, however, does not distinguish between the observed move falling outside the model's leading candidates and being ranked highly but not first. We study this distinction on 884,049 positions from the Allie Lichess 2022 blitz test set using two strong, independently developed population-level human-move predictors, MAIA3 and Allie policy-only, evaluated under the same legal-move protocol. MAIA3 and Allie policy-only rank the observed human move in Top5 on 91.843% and 90.932% of positions, respectively, while reaching only 57.255% and 55.734% Top1 accuracy. We use the term *argmax gap* for this empirical Top5–Top1 discrepancy.

A diagnostic MAIA3–Allie oracle reaches 61.796% Top1, but fixed selectors, tested pre-move learned selector baselines, including linear, small-MLP, and XGBoost baselines, and probability ensembles recover little of this headroom because their rescues are largely offset by newly introduced errors. A conservative rank-2 correction gate achieves a small but reliable gain over MAIA3, improving Top1 by 0.137 percentage points, but most oracle headroom remains unrecovered. The discrepancy persists across chess-context and model-uncertainty strata, while performance varies substantially across observed move-time and clock-context groups. Calibration, refinement, and ensembling improve NLL with little or no corresponding improvement in Top1. Because each position contributes only one observed move, these results do not establish how much of the discrepancy is reducible. They nevertheless show that ranking the observed move highly, fitting its probability well, and matching it exactly at Top1 are distinct evaluation outcomes.

## 1 Introduction

As AI systems become superhuman in games such as chess (Silver et al., 2018; Campbell et al., 2002; Sadmine et al., 2023), an increasingly important challenge is to model the decisions people are likely to make. A chess engine seeks the strongest move, whereas a model of human play seeks the move a person will choose. That choice depends not only on the board but also on factors such as skill, preparation, habits, risk preference, time pressure, and plan. Human move prediction is therefore a distinct problem.

This distinction has motivated a line of work on human chess modeling. MAIA introduced human move prediction as a problem of modeling human play rather than approximating engine play (McIlroy-Young et al., 2020), and MAIA2 extended this approach by conditioning predictions on player skill (Tang et al., 2024). MAIA3 further provides a strong modern population-level human-move policy (Monroe et al., 2026). Allie broadens this task to game-log behavior, such as thinking time (Zhang et al., 2024), while Maia4All shows that stable player histories can capture persistent individual differences beyond rating (Tang et al., 2025). Across this line of work, exact move matching remains central, while related analyses also study skill-conditioned behavior, move-prediction perplexity and coherence, game-log behavior, and player-specific adaptation (McIlroy-Young et al., 2020; Tang et al., 2024; Monroe et al., 2026; Zhang et al., 2024; Tang et al., 2025). These complementary views motivate a closer look at how ranking, probability quality, and final move selection relate in strong current predictors. MAIA3 and Allie are a natural pair for this study because they are strong public policies from distinct modeling lines, produce legal-move distributions, and can be evaluated under the same shared protocol. These developments motivate the question we study: once

a strong model already ranks the observed human move highly, why does exact Top1 matching remain much lower?

We find a large discrepancy between Top5 inclusion and exact move matching. On the shared-protocol Allie blitz evaluation set, MAIA3 ranks the observed human move in Top5 on 91.843% of positions, but reaches only 57.255% Top1 accuracy, where Top1 means exact move matching, i.e., the model's highest-ranked legal move is the move actually played. Allie policy-only also shows the same pattern, with 90.932% Top5 and 55.734% Top1 accuracy. The observed move is therefore usually among the models' five highest-ranked candidates, while exact move matching remains much lower. Figure 1 illustrates this situation: both models rank the human move second, but neither selects it as Top1.

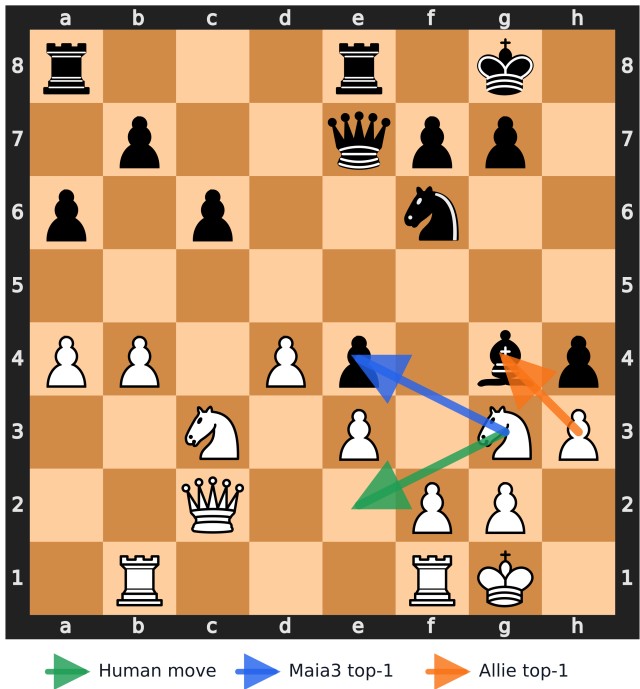

Human move rank: Maia3 #2, Allie #2

Figure 1: A blitz-game position illustrating the argmax gap. The human move is shown in green, while the MAIA3 and Allie top moves are shown in blue and orange. Both models rank the human move second.

We use the term argmax gap to denote the discrepancy between ranking the observed human move highly and making it the highest-ranked prediction. We use this term as a descriptive label for the empirical Top5–Top1 discrepancy studied here, rather than as a new classification metric. We also observe that improving a probability distribution does not necessarily close this gap, because the probability assigned to the human move can increase while another move remains ranked first. To analyze changes in Top1, we use a rescue/break accounting of paired prediction outcomes: an intervention *rescues* a position when it corrects a base-model error and *breaks* a position when it changes a correct base prediction into an error. We observe that relative to MAIA3, Allie policy-only rescues 40,146 errors but breaks 53,592 correct predictions.

The two models also contain complementary information. A diagnostic MAIA3–Allie oracle reaches 61.796% Top1, showing substantial headroom beyond either model alone. Yet fixed selectors, tested pre-move learned selector baselines, including linear, small-MLP, and XGBoost baselines, and held-out-selected probability ensembles recover little of this headroom. A conservative rank-2 correction gate achieves a small but reliable gain over MAIA3, improving Top1 by 0.137 percentage points, but most oracle headroom remains unrecovered. The gap is also not confined to one type of position and persists across game phase, rating,

legal-move count, and model uncertainty strata. Time-related analyses further reveal the structure: both raw move time and the fraction of the remaining clock spent on a move separate easier and harder blitz decisions. Calibration, trust-region refinement, and probability ensembling improve NLL with little or no corresponding improvement in Top1. MRR and NDCG provide a midpoint between NLL and Top1, and show that these probability-level improvements also do not produce substantial intermediate-rank gains. Together, these results distinguish three related evaluation outcomes: ranking the observed move highly, fitting its probability well, and matching it exactly at Top1. Top1 remains the direct metric for exact move prediction, while the additional metrics help diagnose where exact matching succeeds or fails.

Our contributions are the following:

- We empirically characterize the Top5–Top1 discrepancy in strong human-move predictors, which we refer to as the *argmax gap*. We use rescue/break accounting to show how corrected errors and newly introduced errors determine the net change in Top1 accuracy.

- We quantify complementary correct predictions between MAIA3 and Allie and evaluate fixed selectors, linear, small-MLP, and XGBoost learned selector baselines, trust-region refiners, and probability-level ensembles. These methods recover little of the oracle headroom relative to MAIA3, although shortlist reranking improves Allie. We further show that a conservative rank-2 correction gate improves MAIA3 by 0.137 percentage points while switching on 5.338% of positions, recovering a small but reliable fraction of the oracle headroom.

- We show that model performance varies substantially across observed move-time and chess-phase groups. Top1 decreases sharply across longer move-time groups, while it increases and the Top5–Top1 discrepancy narrows across the move-number-based phase groups.

- We show that better probability modeling does not necessarily improve exact move matching. Calibration, trust-region refinement, and probability ensembling improve NLL with little or no corresponding improvement in Top1. MRR and NDCG analyses provide a midpoint check and show that these distributional improvements do not produce substantial intermediate-rank gains.

## 2 Problem Formulation

To make the argmax gap explicit, we treat a human move predictor ($\pi$) as both a probability model and a decision rule. Let the evaluation set contain $N$ positions. For each position $i \in \{1, \ldots, N\}$, let $x_i$ denote the pre-move context, $A_i$ the set of legal moves, and $y_i \in A_i$ the move played by the human. Depending on the model, $x_i$ may include the board position, move history, rating information, or clock-related features. We write $\pi$ for a predictor or policy. When $\pi$ produces normalized probabilities over legal moves, we denote its probability for move $a \in A_i$ by

$$\pi_i(a) = \pi(a \mid x_i, A_i).$$

The probability distribution $\pi_i$ measures how much mass the model assigns to each legal move, while the induced decision rule selects the move with the largest probability.

We evaluate ranked predictions using a tie-aware definition of Top-$k$ accuracy. This is important because legal-move scores can contain ties, and arbitrary tie-breaking can change the apparent Top1 result. For a predictor $\pi$, we define the rank of the human move as

$$\mathrm{rank}_\pi(y_i) = 1 + |\{a \in A_i : \pi_i(a) > \pi_i(y_i)\}|.$$

The human move is counted as Top-$k$ if $\mathrm{rank}_\pi(y_i) \leq k$. Equivalently,

$$\mathrm{Top}k(\pi) = \frac{1}{N} \sum_{i=1}^{N} \mathbf{1}\left[\mathrm{rank}_\pi(y_i) \leq k\right].$$

Thus, Top1 means that no legal move receives strictly greater probability than the human move. In the main text, we use Top5 as the primary shortlist metric because it asks whether the observed human move

appears among the model's five highest-ranked candidates. It does not imply that the remaining Top5 moves constitute the complete set of moves that humans would consider plausible.

As midpoint ranking metrics between Top1 and likelihood, we also use mean reciprocal rank and normalized discounted cumulative gain. Mean reciprocal rank is defined as

$$\text{MRR}(\pi) = \frac{1}{N} \sum_{i=1}^{N} \frac{1}{\text{rank}_\pi(y_i)}.$$

For a single observed relevant move, the ideal discounted gain is 1, so NDCG@$k$ is

$$\text{NDCG@}k(\pi) = \frac{1}{N} \sum_{i=1}^{N} \mathbf{1}\left[\text{rank}_\pi(y_i) \leq k\right] \frac{1}{\log_2(\text{rank}_\pi(y_i) + 1)}.$$

These metrics ask whether the observed move moves closer to the top of the ranked list, even when the exact Top1 decision does not change.

When normalized probabilities are available, we also report negative log-likelihood,

$$\text{NLL}(\pi) = -\frac{1}{N} \sum_{i=1}^{N} \log \pi_i(y_i).$$

Top1 and NLL measure different properties. Top1 evaluates the move selected by the model, while NLL measures how much probability the model assigns to the move actually played. A model can improve NLL by increasing $\pi_i(y_i)$ while another move remains ranked first. In that case, the probability model improves, but exact move matching does not.

We use the term *argmax gap* for this Top-$k$–Top1 discrepancy. For a shortlist size $k$, we write

$$G_k(\pi) = \text{Top}k(\pi) - \text{Top1}(\pi).$$

A large $G_5(\pi)$ means that the observed human move is frequently ranked within Top5 but not as Top1. This is a reporting convention based on standard Top-$k$ and Top1 metrics; it does not by itself determine why the observed move was not ranked first or how much of the discrepancy is reducible. We use $G_k$ descriptively to summarize how often the observed move is close to the top of a model's ranking while a different move is selected as the argmax. This separates exact move matching from shortlist ranking while retaining the standard Top-$k$ and Top1 evaluation definitions. More generally, the argmax gap highlights that improvements in likelihood need not imply improvements in exact move matching. Given a base predictor $\pi_0$ and an alternative predictor $\pi_1$, define

$$\Delta\text{NLL} = \text{NLL}(\pi_1) - \text{NLL}(\pi_0)$$

and

$$\Delta\text{Top1} = \text{Top1}(\pi_1) - \text{Top1}(\pi_0).$$

Then the key point is

$$\Delta\text{NLL} < 0 \;\not\Rightarrow\; \Delta\text{Top1} > 0.$$

This distinction is central to the paper: improving the distribution over human moves is not necessarily the same as improving the argmax decision induced by that distribution.

To analyze changes in Top1, we use a rescue/break accounting of paired prediction outcomes. For any predictor $\pi$, let

$$c_i(\pi) = \mathbf{1}\left[\text{rank}_\pi(y_i) = 1\right]$$

indicate whether $\pi$ predicts the human move as Top1 on position $i$. A *rescue* occurs when the base predictor $\pi_0$ is wrong and the alternative predictor $\pi_1$ is correct,

$$\text{Rescue}_i(\pi_0, \pi_1) = \mathbf{1}\left[c_i(\pi_0) = 0 \land c_i(\pi_1) = 1\right],$$

and a *break* occurs when the base predictor is correct, and the alternative predictor is wrong,

$$\text{Break}_i(\pi_0, \pi_1) = \mathbf{1}\left[c_i(\pi_0) = 1 \wedge c_i(\pi_1) = 0\right].$$

The change in Top1 is exactly

$$\Delta \text{Top1} = \frac{1}{N}\sum_{i=1}^{N}\left(\text{Rescue}_i(\pi_0, \pi_1) - \text{Break}_i(\pi_0, \pi_1)\right).$$

This identity separates corrected base-model errors from previously correct predictions changed into errors, showing how the net Top1 change is produced. A method may rescue many base-model errors, but it improves Top1 only if it rescues more positions than it breaks.

A small example illustrates the accounting. If the observed human move is ranked second, the position is Top5-correct but Top1-wrong, and therefore contributes to $G_5$. If an intervention changes a wrong base-model Top1 prediction to the observed move, it is a rescue. If it changes a previously correct Top1 prediction to another move, it is a break. One rescue and one break cancel in the aggregate Top1 change, which is why reporting only corrected errors can be misleading.

Finally, we use move time as an observed property of the human decision. Let $\tau_i$ denote the time spent on move $y_i$, and let

$$z_i = g(\tau_i)$$

denote its move-time bucket. The two base policies and the primary learned selectors are not conditioned on $\tau_i$ or $z_i$. Their predictions are generated using information available before the target move is played. Afterward, observed move time is used for post-hoc stratification and time-bucket calibration. Since the benchmark consists of blitz games, these buckets should be interpreted as blitz move-time groups. This analysis describes how prediction quality, uncertainty, and argmax errors vary across faster and slower observed decisions. It does not identify a causal effect of move time or isolate it from correlated factors such as game phase, legal-move count, rating, and clock state.

## 3 Experimental Setup

### 3.1 Evaluation Set

We evaluate on a filtered position-level subset of the Allie Lichess 2022 blitz test split (Zhang et al., 2024). Because our main comparison is between MAIA3 and Allie, we retain only positions that both systems can score under the same move-prediction protocol. The target move must align with the pre-move position, the legal move set must be recoverable, and all context fields required by the evaluators must be available. This yields a shared evaluation set of 884,049 positions, on which all main results are reported.

Each row represents one human move decision. Given the pre-move context and legal moves, a model assigns scores or probabilities to the legal moves, and the target is the move actually played. The realized duration of the target move is not an input to either base policy or to the primary learned selectors. It is used only in post-hoc analyses, including performance stratification, time-bucket calibration, and the refiner initialized from that calibration. Because the source games are blitz games, the resulting groups should be interpreted as observed blitz move-time groups rather than as general chess thinking-time categories. Appendix F reports the complete filter flow, checkpoint, and dataset provenance, split-independence checks, and legal-move validation.

### 3.2 Models and Comparisons

We evaluate two strong, independently developed human-move predictors: MAIA3 (79M checkpoint) and Allie policy-only. MAIA3 is our primary population-level reference model, while Allie provides a second policy from a distinct modeling line. Both models are evaluated with identical target moves and legal move sets. These models are a natural pair for our study because they are strong public human-move policies, come

from different modeling lines, produce legal-move distributions needed for rank-based and probability-based evaluation, and can be evaluated under the same shared Allie blitz protocol.

We first test whether MAIA3–Allie complementarity can be used through hard selection. Fixed selectors choose between the two Top1 moves using top-move probability, legal-move entropy, or the Top1–Top2 probability margin. Learned selectors are trained on off-test data and either choose between the two model outputs or rerank candidates from one model's shortlist. The primary learned selectors use 45 features available before the target move is played, covering model confidence, candidate rank and probability, player-rating context, pre-move position context, model agreement, and move geometry. Realized move duration, its logarithm, and its derived move-time bucket are excluded. The primary learned selectors are linear candidate-level models. To test whether the selector results are specific to this model class, we also evaluate small-MLP and XGBoost selector baselines on the same audited 45 pre-move/model-output feature matrix. The full candidate sets, feature groups, and training protocol are given in Appendix B.

We also evaluate conservative break-aware correction gates as direct attempts to reduce the Top5–Top1 discrepancy. These methods keep MAIA3's Top1 prediction by default and switch only when a validation-selected pre-move score predicts a positive net gain. The main such method is a rank-2 correction gate, which considers MAIA3's second-ranked move as the only alternative to the MAIA3 Top1 move. This gate is fit on off-test data, its threshold is selected on validation data, and final evaluation is performed only after these choices are fixed.

We also combine the complete legal-move distributions using convex and geometric probability mixtures (Genest & Zidek, 1986). Mixture weights are selected by held-out NLL and fixed before evaluation on the paper test set. Finally, we report a diagnostic oracle that is counted correct whenever either MAIA3 or Allie predicts the human move as Top1. Because the oracle uses the human move label, it summarizes the union of the models' Top1-correct positions rather than defining a deployable predictor.

We next study calibration (DeGroot & Fienberg, 1983; Guo et al., 2017; Nixon et al., 2019) and a held-out trust-region refiner. Calibration uses temperature scaling fitted separately for the five primary move-time buckets on 500,000 held-out positions by minimizing human-move NLL. For bucket $z_i$,

$$p_i^{\mathrm{cal}}(a) \propto p_i(a)^{1/T_{z_i}}.$$

Since the same positive temperature $T_{z_i}$ is applied to every legal move in the bucket, calibration preserves the legal-move ranking and therefore leaves Top1 unchanged. Because it uses the observed move-time bucket, it is treated as a post-hoc diagnostic rather than a deployable pre-move transformation.

The refiner further applies a bounded residual to candidates from the calibrated distribution:

$$q_i(a) \propto p_i^{\mathrm{cal}}(a) \exp(\alpha r_i(a)).$$

The residual $(r_i)$ is produced by a linear scorer over 29 pre-move confidence, rating, position-context, candidate, and move-geometry features. It is set to zero outside the base model's Top-$K$ candidates, so all other legal moves retain their calibrated probability before global renormalization. The scorer is trained on 400,000 held-out positions and selected on a separate 100,000-position validation split using weighted candidate-level cross-entropy with $L_2$ regularization. We search $K \in \{5, 10, 20\}$, $\alpha \in \{0.05, 0.1, 0.2, 0.5\}$, and three rescue/break weighting profiles. Validation selects $K = 5$, $\alpha = 0.05$, and the rescue-weighted profile for both models. The refinement is constrained operationally through Top-$K$ support, residual clipping, and the small residual scale; it does not optimize an explicit KL constraint. Because the refiner begins from the time-bucket-calibrated distribution, it is also interpreted as a post-hoc diagnostic.

### 3.3   Metrics and Protocol

We report Top1, Top3, Top5, Top10, and Top20 accuracy using the tie-aware rank definition from Section 2. Top1 measures exact move matching, while Top5 records whether the observed human move appears among the model's five highest-ranked candidates. Top5 inclusion does not imply that these candidates constitute the complete set of moves that humans would regard as plausible. Together, the ranking metrics distinguish between ranking the observed move highly and selecting it as the top prediction. We treat these metrics

as complementary diagnostics rather than replacements for Top1: exact move matching remains the direct metric for predicting the move that was actually played, while Top-$k$ and related ranking metrics describe where the observed move appears in the model's ordering.

For probability-level methods, we also report MRR and NDCG@$k$ as intermediate ranking metrics between Top1 and NLL. These metrics use the same strict-greater rank convention from Section 2 and ask whether the observed human move moves closer to the top of the ranked legal-move list.

When normalized probabilities are available, we additionally report negative log-likelihood (NLL). For calibration, we report expected calibration error (ECE) using the probability assigned to the selected Top1 move as confidence and Top1 correctness as the outcome (Naeini et al., 2015). ECE is computed with ten equal-width bins over $[0, 1]$, skipping empty bins and weighting each confidence–accuracy gap by its share of positions. Disagreement-row ECE applies the same calculation only to positions where MAIA3 and Allie select different Top1 moves.

For paired comparisons against a base model, we report rescues, breaks, and net Top1 change. A rescue corrects a base-model error, whereas a break changes a base-correct prediction into an error. Reporting both shows how corrected errors and newly introduced errors combine to produce the net Top1 change.

For the descriptive move-time analyses, we use the primary blitz buckets

$$[0, 1], \quad (1, 2], \quad (2, 4], \quad (4, 7], \quad (7, \infty)$$

seconds. These comparisons report how performance varies across observed move-time groups. An additional fixed-second analysis and clock-context robustness are further provided in Appendix C.

Further, all the learned components and configuration choices are determined without using labels from the paper test set. The primary learned selectors are fitted on off-test data using only pre-move features, while calibration parameters, trust-region configurations, and ensemble weights are selected on held-out data. The small-MLP and XGBoost selector baselines follow the same protocol: model families, hyperparameters, and thresholds are selected on held-out validation data, and paper-test labels are used only after selection for final evaluation. The conservative rank-2 correction gate follows the same separation: its score is fit on off-test data, its switching threshold is selected on validation data, and paper-test labels are used only for the final fixed evaluation. Once these choices are fixed, every method is evaluated on the same 884,049 positions. The resulting Top1 comparisons are therefore paired; Appendix E reports confidence intervals and McNemar tests (McNemar, 1947) for the main deltas.

## 4 Results

### 4.1 Strong Models Rank the Observed Human Move in Top5, but Often Not as Top1

We begin by examining whether the observed human move typically receives a low rank when a model is Top1-wrong, or whether it often remains among the model's leading candidates. This distinction is important for the rest of the paper. If the observed move were usually absent from the highest-ranked candidates, Top5 and Top1 would fail on largely the same positions. If it is frequently ranked within Top5 but not first, the two metrics capture different aspects of model performance. Because each position contributes only one observed human move, this analysis concerns the rank assigned to that move; it does not identify the complete set of moves that humans would consider plausible.

Table 1 reports the full Top-$k$ results for MAIA3 and Allie policy-only on the shared-protocol Allie blitz evaluation set. MAIA3 reaches 57.255% Top1 accuracy, but ranks the observed human move within Top5 on 91.843% of positions. Allie policy-only shows the same pattern, with 55.734% Top1 accuracy and 90.932% Top5 accuracy. Thus, for both models, the observed move is usually among the five highest-ranked candidates, while exact move matching remains much lower.

Figure 2 shows the same pattern across several values of $k$. The curves rise quickly from Top1 to Top5 and then approach saturation at larger $k$. Top10 and Top20 further show that the observed move is rarely far

Table 1: Full-evaluation results for MAIA3 and Allie policy-only on the shared-protocol Allie blitz evaluation set. Both models rank the observed human move within Top5 on over 90% of positions, while exact Top1 remains around 56–57%.

| Model | NLL | Top1 | Top3 | Top5 | Top10 | Top20 | Top5–Top1 |
|---|---|---|---|---|---|---|---|
| MAIA3 | 1.292 | 57.255 | 83.876 | 91.843 | 97.836 | 99.770 | 34.588 pp |
| Allie policy-only | 1.348 | 55.734 | 82.622 | 90.932 | 97.399 | 99.691 | 35.199 pp |

outside the models' highest-ranked candidates. The main discrepancy is already visible at Top5: the Top5–Top1 difference is 34.588 percentage points for MAIA3 and 35.199 percentage points for Allie policy-only.

This establishes the basic empirical form of the argmax gap. Even when the model does not place the observed move first, it often assigns that move a high rank. The result describes a shortlist-to-Top1 discrepancy; it does not establish whether the Top5 candidates form the complete set of human-plausible moves or how much of the discrepancy is reducible through better modeling.

The same pattern appears across different chess contexts. Appendix D reports stratified results by game phase, rating, legal-move count, and model uncertainty. For MAIA3, the Top5–Top1 gap is 36.216 percentage points in openings, 34.447 in middlegames, and 31.049 in endgames; Allie policy-only follows the same trend. The gap also varies with the number of legal moves, from 28.246 percentage points in positions with 1–20 legal moves to 36.421 in positions with 41–60 legal moves. Although the gap narrows in endgames and lower-branching positions, it remains substantial across the well-populated strata.

This pattern also motivates the intervention analyses that follow. A method that increases the probability assigned to the observed move may still leave another move ranked first. Similarly, a method that changes the argmax may correct some base-model errors while changing positions that the base model already predicted correctly. We therefore next examine whether MAIA3 and Allie make complementary correct predictions, and whether the tested methods can use that complementarity to improve Top1.

## 4.2 MAIA3 and Allie Contain Complementary Signal

To answer this, we first examine where the two models are correct and wrong. Figure 3 partitions the evaluation set by the Top1 outcome of MAIA3 and Allie policy-only. The models are both correct on 452,569 positions, or 51.2% of the evaluation set, and both wrong on 337,742 positions, or 38.2%. The remaining positions show that the models make different correct predictions: MAIA3 alone is correct on 53,592 positions, while Allie alone is correct on 40,146 positions.

The off-diagonal cells define the additional Top1 accuracy available to an oracle that can choose between the models using the human move label. Such an oracle is correct whenever either MAIA3 or Allie is correct and reaches 61.796% Top1 accuracy, a 4.541 percentage point lift over MAIA3. It is not a deployable predictor, but it shows that the two policies make complementary correct predictions.

The same accounting also shows what a selector would need to distinguish. The 40,146 Allie-only positions are potential rescues relative to MAIA3: switching from MAIA3 to Allie would correct a MAIA3 error. The 53,592 MAIA3-only positions are potential breaks: the same switch would change a MAIA3-correct prediction into an error. Complementarity alone is therefore not sufficient for a deployable improvement. A selector must identify enough Allie-only positions while avoiding too many MAIA3-only positions. We next test whether the fixed and learned selectors considered in this paper can do this.

## 4.3 Break-Aware Selection Recovers a Small Gain

The previous subsection showed that MAIA3 and Allie make complementary correct predictions. We now ask whether this complementarity can be used without access to the human move label. A natural first approach is to switch between the two models using quantities already available from their predicted distributions. For example, one might choose the model with the higher top-move probability, the lower entropy, or the

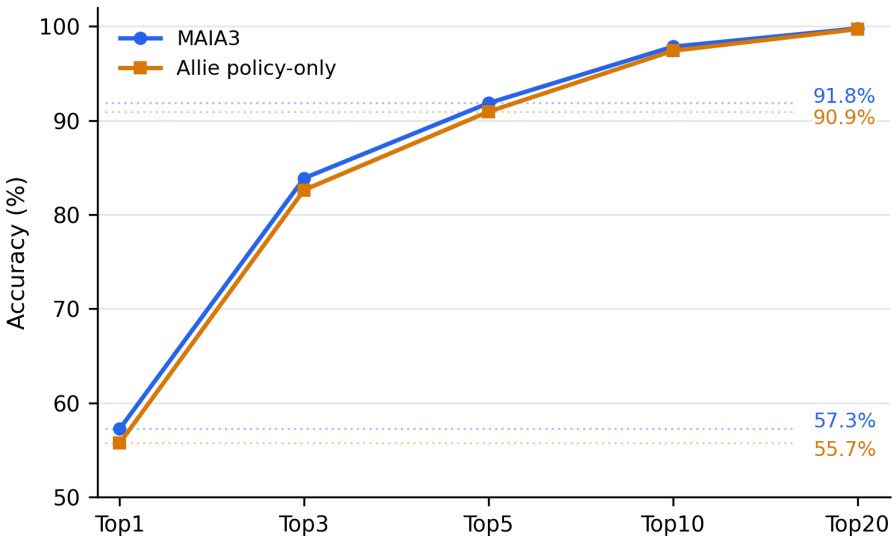

Figure 2: Top-*k* accuracy of MAIA3 and Allie policy-only on the shared-protocol Allie blitz evaluation set. Both models rank the observed human move within Top5 on over 90% of positions, while exact Top1 remains around 56–57%. Dotted reference lines mark the Top1 and Top5 accuracies for each model.

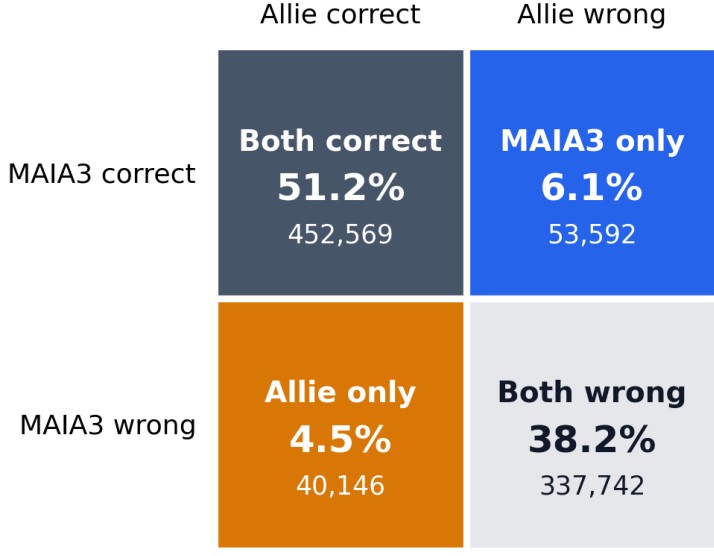

Figure 3: Top1 outcome overlap between MAIA3 and Allie policy-only on the shared-protocol Allie blitz evaluation set. The off-diagonal cells show complementary correct predictions: positions where only MAIA3 is correct or only Allie is correct.

larger gap between its top moves. If these quantities reliably identified which model was more likely to be correct on a disagreement, they should improve over simply using MAIA3 everywhere.

Table 2 shows that the fixed selectors do not improve over MAIA3. The raw max-probability selector reaches 57.039% Top1, below the MAIA3 baseline of 57.255%. Selectors based on entropy and margin also remain below MAIA3, reaching 56.963% and 57.010% Top1, respectively. Thus, these individual confidence summaries do not reliably identify when switching away from MAIA3 will help.

The rescue/break counts show how these net changes are produced. The raw max-probability selector rescues 19,962 positions where MAIA3 is wrong, but breaks 21,874 positions where MAIA3 is already correct, giving

a net change of −1,912 positions. The entropy- and margin-based selectors follow the same pattern: they make genuine corrections, but they also introduce enough new errors for the net Top1 change to remain negative.

We also test whether the selection rule can be learned from off-test data using information available before the target move is played. The cross-model trained selector chooses between the MAIA3 and Allie top moves. It reaches 57.052% Top1, again below MAIA3. It rescues 21,596 MAIA3 errors but breaks 23,392 MAIA3-correct predictions, producing a net change of −1,796 positions. This shows that the tested pre-move linear selector does not convert the available complementarity into a Top1 improvement over MAIA3.

The shortlist selectors test a different question. Instead of choosing between MAIA3 and Allie, a shortlist selector chooses among candidates proposed by a single base model. The MAIA3 shortlist selector asks whether MAIA3's own candidate list can be reranked to improve its default Top1 prediction. It reaches 57.237% Top1, nearly matching MAIA3 but remaining slightly below it. The selector rescues 12,909 MAIA3 errors and breaks 13,067 MAIA3-correct predictions, giving a near-neutral but negative net change of −158 positions.

The Allie shortlist selector provides a useful contrast. It improves Allie policy-only by 1.432 percentage points, reaching 57.165% Top1. It rescues 33,004 Allie errors and breaks 20,348 Allie-correct predictions, for a net gain of 12,656 positions. Learned reranking can therefore improve the weaker Allie policy, although its final Top1 remains 0.089 percentage points below MAIA3.

The conservative rank-2 gate directly targets the rescue/break trade-off. It keeps MAIA3's Top1 prediction by default and switches only to MAIA3's second-ranked move when a validation-selected pre-move score predicts a positive net gain. On the paper test set, it improves MAIA3 from 57.255% to 57.391% Top1. This corresponds to 14,176 rescues and 12,969 breaks, for a net gain of 1,207 positions, while switching on 5.338% of positions. The gain is small but reliable, with both the position-level and 10k game-clustered 95% intervals equal to [+0.100, +0.173] percentage points. It recovers 3.01% of the 4.541 percentage point diagnostic-oracle headroom.

All primary learned-selector results use the 45-feature pre-move specification. Appendix B reports a sensitivity analysis showing that adding realized move duration and its derived features changes Top1 by at most 0.002 percentage points across the three learned selectors. The conclusions are therefore not driven by access to post-decision move time.

To test whether the selector results are specific to linear models, we also evaluate nonlinear selector baselines on the same audited 45-feature pre-move specification. These baselines exclude realized target-move duration and all derived timing features. The small-MLP and XGBoost models are selected on held-out validation data, with paper-test labels used only after selection.

The nonlinear baselines improve some selector variants relative to their linear counterparts, but they do not change the main conclusion. The cross-model MLP and XGBoost selectors remain below MAIA3, and the MAIA3 shortlist XGBoost selector also remains below MAIA3. The Allie shortlist XGBoost selector improves Allie, but still does not exceed MAIA3. Thus, on the audited pre-move feature matrix, tested linear, small-MLP, and XGBoost learned selectors do not recover substantial oracle headroom over MAIA3.

These results refine the selector picture. The fixed, cross-model, and MAIA3-shortlist selectors make genuine corrections, but their breaks offset or exceed their rescues. The Allie shortlist selector improves its own reference model but remains below MAIA3. The conservative rank-2 gate shows that a small amount of headroom is recoverable when changes are made selectively and with an explicit break-control rule. At the same time, the gain remains small relative to the diagnostic oracle, so most complementary signal remains difficult to exploit safely.

## 4.4 Performance Varies Across Observed Move-Time Groups

We next examine a post-hoc property of the game record: the time the player spent on the observed move. Since the evaluation set consists of blitz games, these buckets should be interpreted as observed blitz move-time groups. In this setting, even a few seconds can correspond to a relatively long decision.

Table 2: Selector and break-aware gate outcomes on the shared-protocol Allie blitz evaluation set. Rescue and break counts are measured relative to the reference model shown in the Ref. column.

| Method | Ref. | Top1 | Δ vs. Ref. | Δ vs. MAIA3 | Rescues | Breaks | Net |
|---|---|---|---|---|---|---|---|
| raw max-p | MAIA3 | 57.039 | -0.216 pp | -0.216 pp | 19,962 | 21,874 | -1,912 |
| min-entropy | MAIA3 | 56.963 | -0.292 pp | -0.292 pp | 15,954 | 18,534 | -2,580 |
| max-margin | MAIA3 | 57.010 | -0.245 pp | -0.245 pp | 21,171 | 23,333 | -2,162 |
| cross-model trained | MAIA3 | 57.052 | -0.203 pp | -0.203 pp | 21,596 | 23,392 | -1,796 |
| MAIA3 shortlist selector | MAIA3 | 57.237 | -0.018 pp | -0.018 pp | 12,909 | 13,067 | -158 |
| Allie shortlist selector | Allie | 57.165 | +1.432 pp | -0.089 pp | 33,004 | 20,348 | +12,656 |
| Conservative rank-2 gate | MAIA3 | 57.391 | +0.137 pp | +0.137 pp | 14,176 | 12,969 | +1,207 |

Table 3: Nonlinear selector baselines on the shared-protocol Allie blitz evaluation set. Deltas are reported relative to MAIA3 unless the Ref. column indicates Allie. XGBoost models are selected on held-out validation data and evaluated once on the paper test set.

| Method | Ref. | Top1 | Δ vs. Ref. | Δ vs. MAIA3 | Rescues | Breaks | Net |
|---|---|---|---|---|---|---|---|
| Cross-model MLP selector | MAIA3 | 57.083 | -0.172 pp | -0.172 pp | 16,835 | 18,358 | -1,523 |
| Cross-model XGBoost selector | MAIA3 | 57.097 | -0.158 pp | -0.158 pp | 743 | 2,140 | -1,397 |
| MAIA3 shortlist XGBoost selector | MAIA3 | 57.090 | -0.165 pp | -0.165 pp | 77 | 1,532 | -1,455 |
| Allie shortlist XGBoost selector | Allie | 56.422 | +0.688 pp | -0.833 pp | 10,914 | 4,828 | +6,086 |

Figure 4 shows a strong descriptive association. In the fastest bucket, $[0, 1]$ seconds, MAIA3 reaches 73.405% Top1 accuracy and Allie policy-only reaches 72.596%. In the slowest bucket, $(7, \infty)$ seconds, these values fall to 38.625% and 36.638%, respectively. NLL moves in the opposite direction: MAIA3 NLL rises from 0.812 to 1.876, and Allie NLL rises from 0.837 to 1.967.

Table 4 reports the same pattern numerically. The row percentages show that it is not driven by a small tail bucket: positions are spread across all five move-time groups, with each bucket containing a substantial fraction of the evaluation set. Top1 decreases from bucket to bucket, while NLL increases monotonically for both models.

Move time is also related to the surrounding clock state. Appendix C.2 reports stratification by the remaining clock, whether the time control includes an increment, and the fraction of the pre-move clock spent on the move. The clearest pattern appears for the fraction of clock spent. MAIA3 Top1 decreases from 71.674% when a move uses at most 1% of the available clock to 35.810% when it uses more than 15%; Allie policy-only decreases from 71.075% to 33.196%.

These analyses are descriptive rather than causal. Move time, clock state, move number, game phase, legal-move count, rating, and model uncertainty are related, and the bucket comparisons do not isolate the contribution of any one factor. They nevertheless show that model performance varies substantially across observed move-time and clock-context groups. We next ask whether improving or combining the predicted probability distributions leads to better exact move matching.

## 4.5 Better Probability Models Do Not Necessarily Improve Top1

The move-time results show that prediction performance varies sharply across groups of observed human decisions. We now ask whether improving or combining the predicted probability distributions leads to better exact move matching. A method may assign greater probability to the observed human move without changing which move is ranked first.

Table 5 reports the effects of calibration, trust-region refinement, and probability ensembling. Held-out-fit calibration improves NLL for both models while leaving Top1 unchanged. For MAIA3, NLL improves from 1.292 to 1.284, while Top1 remains 57.255%. For Allie policy-only, NLL improves from 1.348 to 1.338, again

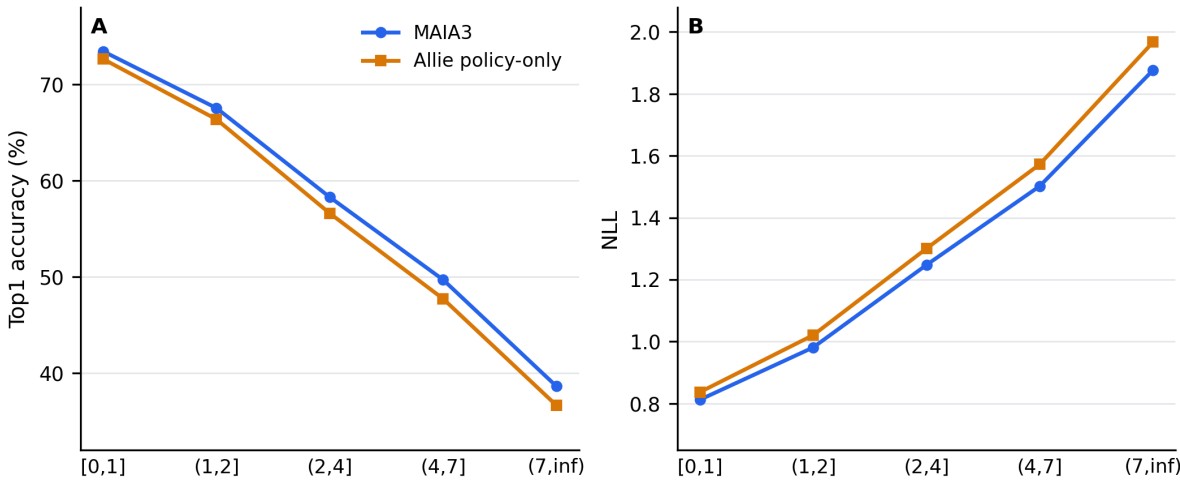

Figure 4: Prediction quality across observed blitz move-time buckets. Longer observed move times are associated with lower Top1 accuracy and higher NLL for both MAIA3 and Allie policy-only.

Table 4: Performance across observed blitz move-time buckets on the shared-protocol Allie blitz evaluation set. The % Rows column is computed from exact bucket counts over all 884,049 positions.

| Bucket | % Rows | MAIA3 Top1 | Allie Top1 | MAIA3 NLL | Allie NLL |
|---|---|---|---|---|---|
| $[0, 1]$ | 20.87 | 73.405 | 72.596 | 0.812 | 0.837 |
| $(1, 2]$ | 18.78 | 67.508 | 66.340 | 0.981 | 1.021 |
| $(2, 4]$ | 20.95 | 58.257 | 56.578 | 1.247 | 1.300 |
| $(4, 7]$ | 16.59 | 49.684 | 47.708 | 1.502 | 1.572 |
| $(7, \infty)$ | 22.81 | 38.625 | 36.638 | 1.876 | 1.967 |

with unchanged Top1. This is expected for rank-preserving calibration: it changes the probability mass assigned to moves but not their ordering.

Trust-region refinement gives the same qualitative result in a less constrained setting. Unlike calibration, it can change the move distribution beyond a rank-preserving rescaling. For MAIA3, the refiner improves NLL relative to the base model, but Top1 decreases from 57.255% to 57.088%. For Allie, the refiner improves NLL while leaving Top1 unchanged. Thus, allowing limited ranking changes is still not sufficient for the tested refiner to improve exact move matching.

We also test whether MAIA3–Allie complementarity can be used by combining their complete legal-move distributions rather than choosing between their top moves. For both convex and geometric mixtures, the MAIA3 weight is selected by held-out NLL and fixed at $\alpha = 0.70$ before evaluation. The convex mixture improves NLL to 1.289 and reaches 57.271% Top1, an increase of only 0.016 percentage points over MAIA3. The geometric mixture improves NLL further, to 1.287, but reaches 57.247% Top1, slightly below MAIA3. Probability ensembling therefore improves distributional fit but recovers essentially none of the 4.541 percentage point diagnostic-oracle difference. Appendix D reports the full mixture-weight sweep.

As a midpoint between NLL and Top1, we also compute MRR and NDCG@$k$ using the same strict-rank convention as the Top-$k$ metrics. These metrics ask whether probability-level modifications move the observed human move closer to the top of the ranking, even when the Top1 decision does not change. Table 6 reports these intermediate ranking metrics.

The midpoint ranking metrics reinforce the main conclusion. Calibration improves NLL but leaves MRR and NDCG exactly unchanged because it is rank-preserving. MAIA3 trust-region refinement improves NLL but slightly worsens Top1, MRR, and NDCG. Convex and geometric ensembles give tiny MRR/NDCG

Table 5: Probability-level modifications improve NLL but yield little or no Top1 improvement. Calibration is rank-preserving, while trust-region refinement and probability ensembling can alter the distribution without producing a meaningful Top1 gain. Deltas for the ensemble rows are measured relative to MAIA3.

| Model | Method | NLL | Top1 | $\Delta$ NLL | $\Delta$ Top1 |
|---|---|---|---|---|---|
| MAIA3 | base | 1.292 | 57.255 | – | – |
| MAIA3 | calibrated | 1.284 | 57.255 | -0.008 | +0.000 pp |
| MAIA3 | trust-region | 1.285 | 57.088 | -0.007 | -0.167 pp |
| Allie | base | 1.348 | 55.734 | – | – |
| Allie | calibrated | 1.338 | 55.734 | -0.010 | +0.000 pp |
| Allie | trust-region | 1.340 | 55.734 | -0.009 | +0.000 pp |
| MAIA3+Allie | convex ensemble ($\alpha = 0.70$) | 1.289 | 57.271 | -0.004 | +0.016 pp |
| MAIA3+Allie | geometric ensemble ($\alpha = 0.70$) | 1.287 | 57.247 | -0.005 | -0.008 pp |

Table 6: Intermediate ranking metrics for probability-level methods. Top1 and Top5 are percentages. MRR and NDCG use the same strict-greater rank convention as the main Top-$k$ metrics.

| Method | NLL | Top1 | Top5 | MRR | NDCG@5 | NDCG@10 |
|---|---|---|---|---|---|---|
| MAIA3 base | 1.292239 | 57.255 | 91.843 | 0.719713 | 0.762304 | 0.782034 |
| MAIA3 calibrated | 1.284101 | 57.255 | 91.843 | 0.719713 | 0.762304 | 0.782034 |
| MAIA3 trust-region | 1.285063 | 57.088 | 91.791 | 0.718551 | 0.761235 | 0.781153 |
| Allie base | 1.348264 | 55.734 | 90.932 | 0.707164 | 0.749878 | 0.771160 |
| Allie calibrated | 1.338168 | 55.734 | 90.932 | 0.707164 | 0.749878 | 0.771160 |
| Allie trust-region | 1.339608 | 55.734 | 90.931 | 0.707163 | 0.749871 | 0.771159 |
| Convex ensemble | 1.288666 | 57.271 | 91.837 | 0.719819 | 0.762361 | 0.782091 |
| Geometric ensemble | 1.287252 | 57.247 | 91.852 | 0.719719 | 0.762333 | 0.782069 |

point gains, but their confidence intervals include zero, and their Top1 changes remain negligible. Thus, probability-level improvements do not reliably translate into either hard Top1 recovery or substantial intermediate-rank gains.

The disagreement-row calibration analysis provides additional context for confidence-based selection. Table 7 compares expected calibration error on all rows and on rows where MAIA3 and Allie disagree in their Top1 predictions. Both models have global ECE below one percentage point. On disagreement rows, however, ECE rises to 2.796 pp for MAIA3 and 9.746 pp for Allie. These are the positions on which a cross-model selector must decide which prediction to retain.

The calibration, refinement, and ensemble results show that improving NLL does not necessarily produce a corresponding improvement in Top1. Calibration improves NLL while preserving the ranking; trust-region refinement permits limited ranking changes but lowers or preserves Top1; and probability ensembling improves NLL while leaving Top1 essentially unchanged. These results distinguish improvements in the predicted legal-move distribution from improvements in the induced Top1 decision.

All Top1 comparisons are paired on the same 884,049 positions. Appendix E reports confidence intervals and McNemar-style tests for the main deltas. We return to the broader implications of these results in the discussion.

## 5 Discussion

The main implication of our results is that strong human-move predictors often rank the observed human move highly, but are much less reliable at placing it first. This does not imply that every Top1 mismatch is a correctable selection error, or that the models' Top5 candidates constitute the complete set of human-plausible moves. It does show that ranking the observed move highly, assigning it probability, and matching it exactly at Top1 capture different aspects of model performance. Top-$k$ accuracy, NLL, calibra-

Table 7: Expected calibration error on all rows and on MAIA3–Allie Top1-disagreement rows. Calibration error is substantially larger on the rows where cross-model selection is required.

| Model | All rows ECE | Disagreement-row ECE |
|---|---|---|
| MAIA3 | 0.278 pp | 2.796 pp |
| Allie policy-only | 0.590 pp | 9.746 pp |

tion, MRR/NDCG, and Top1 should consequently be read together rather than treated as interchangeable measures of progress. In this view, Top1 remains the direct metric for exact move prediction, while the other metrics help diagnose where exact matching succeeds or fails.

The MAIA3–Allie comparison makes this distinction concrete. The two models have complementary correct predictions, so there is measurable headroom beyond either model alone. Fixed selectors, tested linear, small-MLP, and XGBoost pre-move learned selector baselines, and probability-level ensembles recover little of this headroom relative to MAIA3. These methods correct some MAIA3 errors, but they also change enough MAIA3-correct predictions into errors that their net Top1 change is non-positive or negligible. Removing realized move duration and its derived features changes selector Top1 by at most 0.002 percentage points, showing that this pattern is not driven by post-decision timing information. The rescue/break accounting makes the trade-off transparent: a net gain occurs only when corrections outnumber the previously correct predictions changed into errors. The conservative rank-2 correction gate follows this principle more directly. By keeping MAIA3's Top1 prediction by default and switching only to MAIA3's second-ranked move under a validation-selected rule, it recovers a small but reliable Top1 gain. This shows that some headroom can be recovered, but useful corrections require strong control of the break rate.

The move-time and stratified analyses show that model performance is not uniform across positions. Top1 is substantially higher, and NLL lower, in faster observed move-time groups than in longer move-time groups. The Top5–Top1 gap also persists across game phase, rating, legal-move count, and model-uncertainty groups, while performance varies with the remaining clock and, most clearly, with the fraction of the available clock spent on the move. These are unadjusted descriptive associations within blitz. Move time is related to game phase, legal-move count, rating, clock state, and other aspects of position difficulty, so the results should be read as stratified performance summaries rather than causal estimates of move-time effects. Calibration, trust-region refinement, and probability ensembling provide a separate but related result: the predicted distribution can improve without a corresponding improvement in Top1. The MRR and NDCG analyses further show that these probability-level improvements do not necessarily produce substantial intermediate-rank gains.

Because each position contributes only one observed human move, the argmax gap should be interpreted as a descriptive Top5–Top1 discrepancy. The present evaluation cannot determine how much of the gap reflects correctable model-ranking error, how much results from unobserved player or session context, and how much reflects variation in human choice. Exact Top1 matching is still meaningful: when the task is to predict the single move that was played, it is the most direct metric. At the same time, a single observed move is only one sample from a context-dependent human decision process, so Top1 alone is incomplete as a summary of human-move modeling quality. The benchmark includes rating and clock information, but it does not expose stable player histories or directly represent factors such as preparation, risk preference, habitual style, and the player's current plan. Maia4All shows that stable player histories can help model such differences when they are available (Tang et al., 2025). Future work can therefore combine richer player and session context with stronger selection methods, while also measuring how much of the observed discrepancy is reducible. The break-aware results suggest one direction for such work: future methods may need objectives or post-processing rules that combine distributional quality with decision-aware constraints on when a strong base model's Top1 prediction should be changed.

More broadly, our study suggests that human-move prediction should be evaluated not only by exact Top1, but also by Top-$k$ ranking, likelihood, calibration, MRR/NDCG, and rescue/break behavior.

## 6 Conclusion

Our results identify a consistent discrepancy between ranking the observed human move within Top5 and matching it exactly at Top1. MAIA3 and Allie rank the observed move in Top5 on over 90% of positions, yet exact Top1 remains much lower. Their predictions also contain complementary information: a diagnostic oracle exceeds either model alone, but the tested fixed selectors, linear, small-MLP, and XGBoost pre-move learned selector baselines, and probability-level ensembles recover little of this oracle headroom relative to MAIA3. A conservative rank-2 correction gate achieves a small but reliable gain, improving MAIA3 by 0.137 percentage points, but most oracle headroom remains unrecovered. The rescue/break accounting shows how these net changes arise. Interventions can correct base-model errors, but they improve Top1 only when those corrections are not offset by previously correct predictions changed into errors.

The additional analyses show that the observed discrepancy is not confined to a narrow part of the benchmark. It persists across game phase, rating, legal-move count, and model-uncertainty strata, although its magnitude varies across these settings. Performance also varies substantially across observed move-time and clock-context groups; these patterns are descriptive and do not establish a causal effect of move time. Calibration, trust-region refinement, and probability ensembling improve the predicted distribution, but produce little or no improvement in the final Top1 decision. MRR and NDCG provide the same qualitative message: probability-level improvements do not produce substantial intermediate-rank gains. Human-move prediction should therefore be evaluated as both a ranking problem and a probabilistic forecasting problem: Top-$k$ ranking, likelihood, calibration, MRR/NDCG, and rescue/break behavior reveal information that Top1 alone does not. Top1 remains the most direct metric when the goal is exact move prediction, while the additional metrics serve as diagnostic complements that show where exact matching succeeds or fails.

Our study focuses on population-level prediction in blitz games and on MAIA3 and Allie policy-only, the two strong policies evaluated at full scale under the same legal-move protocol. Since each position provides only one observed human move, the experiments do not establish what fraction of the Top5–Top1 discrepancy is reducible by better modeling, nor whether the models' Top5 candidates represent the complete set of moves humans would consider plausible. The observed discrepancy is therefore a diagnostic property of these models and data, not a bound on achievable Top1 accuracy; stronger models, richer player and session context, and different training or reranking objectives may further improve exact move matching. Future work can address this question through repeated-position analyses, richer player and session histories, additional time controls, and stronger pre-move selection methods. The rank-2 correction gate suggests one useful direction: break-aware objectives or post-processing rules that change a strong base model's prediction only when the expected correction outweighs the risk of introducing a break. A central open problem is to distinguish correctable ranking errors from unobserved or intrinsic variation in human choice while making candidate changes with a sufficiently low break rate.

### Broader Impact

Human-move prediction models may be useful for chess education, training tools, and analysis systems. At the same time, systems that adapt to individual players can involve sensitive behavioral data, including move choices, timing patterns, ratings, and historical play. Such systems should use appropriate privacy protections, transparency, and data-minimization practices, especially when player-specific profiles are constructed or stored.

While move-prediction models may provide contextual signals for cheating, plagiarism, or other misconduct investigations, they should not be treated as standalone evidence. High-stakes judgments should require independent evidence, careful human review, and an opportunity for affected users to contest the decision.

## Generative AI Disclosure

The authors use ChatGPT for language editing and formatting. They also use the Stanford Agentic Reviewer[1] and the CMU Paper Reviewer (Kim et al., 2026) to obtain preliminary feedback and improve the paper. All technical content, experimental design, analyses, and conclusions remain the sole responsibility of the authors.

## Data and Code Availability

The source Allie Lichess 2022 blitz data and the pretrained MAIA3 and Allie checkpoints are publicly available through their original releases. Appendix F documents the construction of the shared-protocol evaluation set and the development–evaluation split checks. All code, processed data, evaluation scripts, and derived analysis artifacts will be made publicly available upon acceptance.

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

# A   Additional Rank and Probability Geometry

The main text reports a large Top5–Top1 discrepancy for both MAIA3 and Allie policy-only. To examine what lies behind this discrepancy, we restrict attention here to positions on which each model is Top1-wrong and ask where the observed human move appears in the remaining ranking. This separates errors in which the observed move is far from the top of the ranking from errors in which it remains highly ranked but is outranked by another legal move. Because each position contributes only one observed move, this analysis characterizes the rank and probability assigned to that move rather than the complete set of moves humans would consider plausible.

Table 8 shows that the latter case is common. MAIA3 is wrong on 377,888 positions, or 42.745% of the evaluation set. On these rows, the observed human move is ranked second in 42.313% of cases, lies within Top5 in 80.917%, and lies within Top10 in 94.937%. Allie policy-only shows a similar pattern: it is wrong on 391,334 positions, with the observed move ranked second in 40.951% of wrong rows, within Top5 in 79.516%, and within Top10 in 94.124%.

Table 8:   Rank geometry on Top1-wrong positions. Percentages are computed within each model's Top1-wrong rows.

| Model | Wrong rows | Rank 2 (%) | Rank $\leq$ 3 (%) | Rank $\leq$ 5 (%) | Rank $\leq$ 10 (%) |
|---|---|---|---|---|---|
| MAIA3 | 377,888 | 42.313 | 62.279 | 80.917 | 94.937 |
| Allie policy-only | 391,334 | 40.951 | 60.742 | 79.516 | 94.124 |

Figure 5 gives a cumulative view of the same pattern. The observed human move is already ranked second on roughly 41–42% of Top1-wrong rows, and the cumulative share rises to about 80% by rank 5. Thus, even after conditioning on a Top1 error, both models usually place the observed move among a small number of leading candidates.

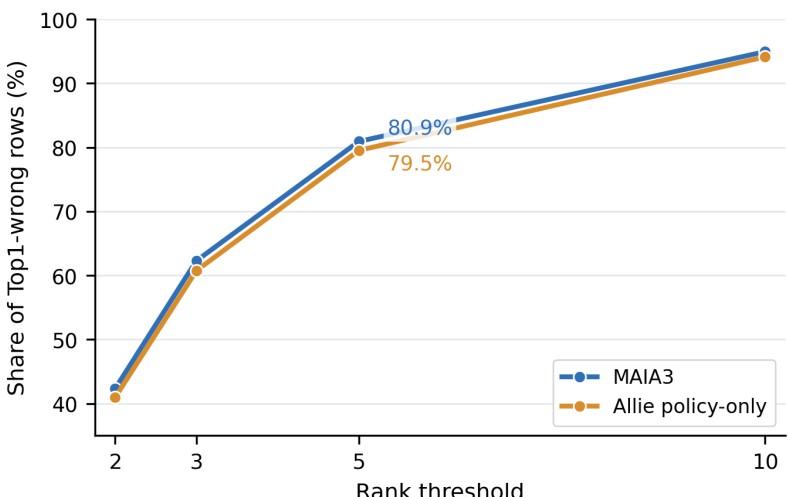

Figure 5:   Cumulative rank of the observed human move among Top1-wrong positions. Even when the argmax is wrong, the observed move remains within Top5 on roughly 80% of wrong rows for both MAIA3 and Allie policy-only.

Proximity in rank does not necessarily mean that the observed move is tied, or nearly tied, with the top prediction. Rank-2 cases with a top1–human probability margin below 0.03 account for only 4.275% of MAIA3 wrong rows and 4.536% of Allie wrong rows. The median margin is 0.264 for MAIA3 and 0.262 for Allie policy-only, showing that many errors retain a meaningful probability separation between the model's top move and the observed move.

Table 9 summarizes the corresponding probability statistics. On Top1-wrong rows, the models assign non-trivial probability to the observed move, but substantially more probability to their top prediction. The observed move is therefore often highly ranked and receives meaningful probability mass without being sufficiently preferred to become the argmax.

Table 9: Probability geometry on Top1-wrong positions. Margins are computed as the top-move probability minus the observed human-move probability.

| Model | Mean $p_{\text{human}}$ | Mean $p_{\text{top1}}$ | Mean entropy | Median margin | 90th pct. margin |
|---|---|---|---|---|---|
| MAIA3 | 0.146 | 0.446 | 1.633 | 0.264 | 0.597 |
| Allie policy-only | 0.142 | 0.441 | 1.672 | 0.262 | 0.596 |

The wrong-row analysis shows that the high Top5 values are not driven only by positions that the models already predict correctly at Top1. Even when the argmax is wrong, the observed human move is usually near the top of the ranked list. This is the setting targeted by the reranking and selection interventions studied in the paper. However, rank and probability geometry alone do not determine whether these Top1 mismatches are correctable or instead reflect unobserved context or variation in human choice.

## B    Complementarity and Selector Details

The preceding wrong-row analysis showed that the observed human move often remains near the top even when the model's argmax is wrong. We therefore consider two ways of changing the final prediction. Cross-model selection chooses between the top moves proposed by MAIA3 and Allie policy-only. Within-model selection instead reranks candidates from a single model's shortlist. Here, a shortlist refers only to the model's highest-ranked candidates; it is not assumed to represent the complete set of moves that humans would consider plausible.

For the cross-model setting, let $\hat{y}_i^M$ and $\hat{y}_i^A$ denote the MAIA3 and Allie Top1 moves on position $i$. Each position belongs to one of four outcome categories: both models are correct, both are wrong, only MAIA3 is correct, or only Allie is correct. The two off-diagonal categories determine the opportunity and risk of switching away from MAIA3. An Allie-only position is a potential rescue, whereas a MAIA3-only position is a potential break. The diagnostic oracle is correct whenever either model is correct,

$$c_i^{\text{oracle}} = \max\{c_i(\text{MAIA3}), c_i(\text{Allie})\}.$$

Because this oracle uses the observed human move label, it summarizes the union of the two models' Top1-correct positions rather than defining a deployable selection rule.

The fixed selectors replace the label-dependent oracle decision with a single statistic derived from the two predicted distributions. For model $m \in \{M, A\}$, let $p_i^{m,(1)}$ and $p_i^{m,(2)}$ denote its largest and second-largest legal-move probabilities. We define its legal-move entropy and Top1–Top2 margin as

$$H_i^m = -\sum_{a \in A_i} \pi_i^m(a) \log \pi_i^m(a), \qquad \gamma_i^m = p_i^{m,(1)} - p_i^{m,(2)}.$$

The three fixed selectors choose the model according to

$$s_i^{\text{max-p}} = \arg \max_{m \in \{M,A\}} p_i^{m,(1)},$$

$$s_i^{\text{min-entropy}} = \arg \min_{m \in \{M,A\}} H_i^m, \qquad s_i^{\text{max-margin}} = \arg \max_{m \in \{M,A\}} \gamma_i^m.$$

The selected prediction is the Top1 move of the chosen model. Ties are resolved in favor of MAIA3. These selectors use only quantities available from the model outputs and do not use the observed human move label.

The primary linear learned selectors replace these fixed rules with a candidate-level scorer. For a candidate move $a$ on position $i$, let $\phi_i(a) \in \mathbb{R}^{45}$ denote its pre-move feature vector. The scorer takes the form

$$s_\theta(i, a) = \theta^\top \phi_i(a), \qquad \hat{y}_i = \arg\max_{a \in C_i} s_\theta(i, a),$$

where $C_i$ is the candidate set for the corresponding selector. The 45 retained features are all available before the target move is played. They cover player-rating context, pre-move position context, MAIA3 and Allie distribution summaries, model agreement and cross-model rank summaries, candidate UCI geometry, and candidate rank and probability under the source and alternative models. Realized move duration, its logarithm, and its derived move-time bucket are excluded from the primary specification.

The primary linear scorer is implemented as an $L_2$-regularized NumPy logistic-regression model trained to distinguish the played move from the alternative candidates. All feature standardization parameters are estimated using the selector-training split only.

The cross-model trained selector uses

$$C_i^{\mathrm{cross}} = \left\{ \hat{y}_i^M, \hat{y}_i^A \right\},$$

and therefore chooses between the two base-model Top1 moves. The shortlist selectors operate within one model instead. The MAIA3 shortlist selector uses MAIA3's Top10 candidates, while the Allie shortlist selector uses Allie's Top5 candidates. Cross-model selection thus asks which policy should be retained on a disagreement; shortlist selection asks whether a policy's own model-defined candidate list can be reranked more effectively than its default argmax. The conservative rank-2 correction gate tests a more restricted intervention: it keeps MAIA3's Top1 prediction by default and considers only MAIA3's second-ranked move as the possible replacement.

Table 10: Selector definitions and reference models. Fixed selectors use a single summary of the predicted distributions, while learned selectors use the candidate-level scorer trained on off-test data using pre-move features only. The rank-2 gate is a conservative break-aware correction rule.

| Method | Candidate set | Decision rule | Ref. |
| --- | --- | --- | --- |
| raw max-p | MAIA3 Top1, Allie Top1 | Choose the larger top-move probability | MAIA3 |
| min-entropy | MAIA3 Top1, Allie Top1 | Choose the lower legal-move entropy | MAIA3 |
| max-margin | MAIA3 Top1, Allie Top1 | Choose the larger Top1–Top2 probability margin | MAIA3 |
| cross-model trained | MAIA3 Top1, Allie Top1 | Candidate-level linear logistic-regression score | MAIA3 |
| MAIA3 shortlist | MAIA3 Top10 | Candidate-level linear logistic-regression score | MAIA3 |
| Allie shortlist | Allie Top5 | Candidate-level linear logistic-regression score | Allie |
| rank-2 correction gate | MAIA3 Top1, MAIA3 rank 2 | Keep MAIA3 Top1 unless a validation-selected pre-move score favors rank 2 | MAIA3 |

The learned selectors are fit on the off-test training split from the held-out 5M data and selected on a separate selector-validation split. The selected logistic-regression configuration uses an $L_2$ coefficient of $10^{-4}$ and a learning rate of 0.05. Configurations are ranked by validation Top1; ties are resolved first by the larger net rescue count and then by the smaller number of breaks. Labels from the final evaluation set are not used for fitting or configuration selection.

**Conservative rank-2 correction gate.** To test whether the rescue/break accounting can guide an improvement rather than only diagnose failures, we evaluate a conservative rank-2 correction gate. The gate uses MAIA3 as the reference policy and keeps MAIA3's Top1 move by default. The only possible switch is to MAIA3's second-ranked move. This design is motivated by the wrong-row rank analysis: many Top1 errors have the observed move near the top of MAIA3's ranking, but unrestricted changes can introduce too many breaks.

Table 11: Training and selection protocol for the primary linear learned selectors.

| Component | Setting |
| --- | --- |
| Model class | Candidate-level NumPy logistic regression |
| Feature dimension | 45 pre-move features |
| Fit data | Off-test selector-training split |
| Selection data | Separate selector-validation split |
| Standardization | Fit on selector-training data only |
| Regularization | $L_2$ coefficient $10^{-4}$ |
| Learning rate | 0.05 |
| Primary selection criterion | Validation Top1 |
| First tie-break | Larger rescues minus breaks |
| Second tie-break | Fewer breaks |
| Evaluation labels used for selection | No |

The rank-2 gate is trained on off-test data using pre-move/model-output features and excludes realized target-move duration, its logarithm, and its derived move-time bucket. The selected model is a ridge-style linear scorer with $L_2 = 0.01$. Its switching threshold is selected on the selector-validation split by validation Top1, with ties resolved by larger net rescues minus breaks and then by fewer breaks. Paper-test labels are used only after the model and threshold are fixed. The selected gate switches from MAIA3 Top1 to MAIA3 rank 2 only when the learned score exceeds the validation-selected threshold.

Table 12: Training and selection protocol for the conservative rank-2 correction gate.

| Component | Setting |
| --- | --- |
| Reference policy | MAIA3 |
| Default prediction | MAIA3 Top1 |
| Alternative candidate | MAIA3 rank 2 |
| Model class | Ridge-style linear scorer |
| Selected regularization | $L_2 = 0.01$ |
| Feature set | Audited pre-move/model-output features; no realized target-move duration |
| Fit data | Off-test selector-training split |
| Selection data | Separate selector-validation split |
| Selection criterion | Validation Top1, then larger net rescue count, then fewer breaks |
| Evaluation labels used for selection | No |

The rank-2 gate search considered ridge models with $L_2 \in \{0.01, 0.1, 1\}$ and one-vs-rest logistic delta models with $L_2 \in \{10^{-4}, 10^{-3}\}$, learning rate 0.05, and three epochs. The selected model is `single_rank2_ridge_l2_0.01`. It uses 65 audited pre-move/model-output features: the shared 45 selector features plus 20 candidate-rank, MAIA3/Allie probability/rank, Top20-mass, and MAIA3 Top1 move-geometry features. Realized target-move duration and derived timing features are excluded. Training standardization is fit on rank-2 candidate rows from the selector-training split only.

The selected switching threshold is $-0.00212198495865$. The exact decision rule is to retain MAIA3 Top1 by default and switch to MAIA3 rank 2 only if the standardized ridge score is strictly greater than this threshold; equality does not switch. The metadata seed is 20260701. The selected ridge fit is closed-form and deterministic, so it does not consume an RNG. On selector validation, the selected gate reaches 56.119% Top1, a +0.178 percentage-point change relative to MAIA3, with 1,577 rescues and 1,399 breaks. On the paper test set, it reaches 57.391% Top1, a +0.137 percentage-point change, with 14,176 rescues, 12,969 breaks, and a 5.338% switch rate.

**Nonlinear selector baselines.** To test whether the selector conclusions are specific to the linear model class, we also evaluate nonlinear selector baselines on the same audited 45-feature pre-move/model-output feature matrix. These baselines use the same train/validation/evaluation separation as the primary learned selectors. Model classes, hyperparameters, and thresholds are selected on the selector-validation split only,

and labels from the final evaluation set are used only after selection for final evaluation. All nonlinear baselines exclude realized target-move duration, its logarithm, and its derived move-time bucket.

We evaluate deterministic small-MLP baselines and XGBoost baselines. The XGBoost experiments use XGBoost 3.2.0 with train-only preprocessing and validation-selected hyperparameters. These nonlinear baselines are evaluated on the same selector tasks as the primary learned selectors: cross-model selection, MAIA3 shortlist reranking, and Allie shortlist reranking.

Table 13: Protocol for nonlinear selector baselines.

| Baseline family | Model class | Selection and evaluation protocol |
|---|---|---|
| Small MLP | Feed-forward neural network over the audited 45 pre-move/model-output features | Architecture and regularization selected on selector-validation data; final evaluation performed once on the paper test set |
| XGBoost | Gradient-boosted tree model using XGBoost 3.2.0 | Hyperparameters selected on selector-validation data; final evaluation performed once on the paper test set |

For the small-MLP baseline, we search hidden-layer widths $(32)$, $(64)$, $(128)$, $(64, 32)$, and $(128, 64)$, with ReLU activations after each hidden layer. The regularization settings are $(\text{dropout} = 0, \text{weight decay} = 0)$, $(\text{dropout} = 0.1, \text{weight decay} = 10^{-4})$, and $(\text{dropout} = 0.1, \text{weight decay} = 10^{-3})$. The loss is class-weighted binary cross-entropy with logits, using a positive weight equal to the sampled negative/positive ratio. The optimizer is AdamW with learning rate $10^{-3}$, batch size 8192, and maximum four epochs. The checkpoint with the smallest class-weighted validation loss is restored after all four epochs; there is no patience-based early stopping. The base seed is 20260711. The selected MLP configuration for the cross-model selector is `cross_model_mlp_h64x32_d0p1_wd1em03`, with hidden layers $(64, 32)$, dropout 0.1, and weight decay $10^{-3}$. It scores the MAIA3 Top1 and Allie Top1 candidates independently and selects the candidate with the larger sigmoid score; no separate threshold is used.

For XGBoost, we use XGBoost 3.2.0 with objective `binary:logistic` and `tree_method=hist`. The search space is $n_{\text{estimators}} \in \{100, 300\}$, maximum depth in $\{2, 3, 4\}$, learning rate in $\{0.03, 0.1\}$, subsample in $\{0.8, 1.0\}$, column subsample in $\{0.8, 1.0\}$, $L_2$ regularization $\lambda \in \{1, 5, 10\}$, and minimum child weight in $\{1, 10\}$, with random state 20260711 and 8 jobs. No evaluation set is passed to the XGBoost fit call, so no evaluation metric controls model selection or stopping; early stopping is not used. The $L_1$ regularization and maximum-bin parameters use XGBoost defaults.

If the full candidate-level training set exceeds the deterministic cap of 80,000 candidate rows, positives are retained when there are at most 40,000 positives; otherwise, 40,000 positives are sampled deterministically and the remaining capacity is filled with nonpositive rows. Validation and paper-test candidate sets are evaluated in full. Selection uses validation Top1, with ties resolved by larger rescues-minus-breaks, fewer breaks, lower change rate, and then simpler/fewer-tree models. The XGBoost decision rule uses `XGBClassifier.predict`, producing hard class predictions for each candidate; the selected move is the candidate with the largest predicted class, with ties resolved by ascending candidate slot, so the reference Top1 candidate is preferred in ties. No separate selector threshold is used for the XGBoost candidate selectors.

Rescues and breaks are measured relative to the reference model shown in Table 10. For the fixed selectors, the cross-model trained selector, and the MAIA3 shortlist selector, a rescue corrects a MAIA3 error and a break changes a MAIA3-correct prediction into an error. For the Allie shortlist selector, both quantities are measured relative to Allie because the selector reranks Allie's own candidates. For the rank-2 correction gate, rescues and breaks are measured relative to MAIA3 because the gate only changes MAIA3's prediction when it switches from rank 1 to rank 2. This rescue/break accounting separates corrected base-model errors from previously correct predictions changed into errors, showing how each net Top1 change is produced.

**Realized-time sensitivity.** An audit of the original cached selector matrix identified three post-decision timing features: realized move duration, its logarithm, and its derived move-time bucket. We therefore

Table 14: Selected nonlinear selector configurations. All models are selected on selector-validation data and evaluated once on the paper test set.

| Task | Selected configuration | Val. Top1 | Paper Top1 | Δ vs. MAIA3 |
|---|---|---|---|---|
| Cross-model MLP | $(64, 32)$ ReLU MLP; dropout 0.1; weight decay $10^{-3}$; AdamW lr $10^{-3}$; batch 8192; four epochs | 55.938 | 57.083 | -0.172 pp |
| Cross-model XGBoost | 300 trees; depth 3; lr 0.1; subsample 0.8; columns 0.8; $\lambda = 5$; min child weight 1 | 55.812 | 57.097 | -0.158 pp |
| MAIA3 Top10 XGBoost | 300 trees; depth 4; lr 0.1; subsample 0.8; columns 1.0; $\lambda = 1$; min child weight 10 | 55.799 | 57.090 | -0.165 pp |
| Allie Top5 XGBoost | 100 trees; depth 3; lr 0.03; subsample 1.0; columns 0.8; $\lambda = 5$; min child weight 1 | 55.432 | 56.422 | -0.833 pp |

exclude these features from the primary selector specification and retrain all learned selectors using the remaining 45 pre-move features. The cached selector matrix contains no pre-move clock-state or time-control features, so additionally removing all clock-related inputs produces the same specification.

Table 15 compares the primary pre-move selectors with otherwise identical diagnostic variants that include the three realized-duration features. Removing these features changes Top1 by at most 0.002 percentage points. In particular, the Allie shortlist selector retains its 1.432 percentage point improvement over Allie policy-only. All main-text results nevertheless use the pre-move-only specification.

Table 15: Sensitivity to realized move-duration features. The difference is the pre-move Top1 value minus the value obtained when realized-duration features are included. All main results use the pre-move specification.

| Selector | Pre-move Top1 | With duration | Difference |
|---|---|---|---|
| cross-model trained | 57.052 | 57.051 | +0.001 pp |
| MAIA3 shortlist | 57.237 | 57.239 | -0.002 pp |
| Allie shortlist | 57.165 | 57.165 | +0.000 pp |

All selectors and the conservative rank-2 gate produce a single hard move choice rather than a normalized legal-move distribution. They are therefore evaluated using Top1 accuracy and the rescue/break accounting; probability-based metrics such as NLL are reserved for calibration, refinement, and ensemble methods that produce complete distributions.

# C   Move-Time Analyses

## C.1   Fixed-Second Robustness

The main analysis groups moves into the primary blitz buckets $[0, 1]$, $(1, 2]$, $(2, 4]$, $(4, 7]$, and $(7, \infty)$ seconds. These boundaries reflect the compressed time scale of blitz. To check that the observed pattern does not depend on this particular choice of buckets, we repeat the analysis using fixed-second intervals.

Figure 6 shows the resulting Top1 and NLL curves. The pattern is unchanged: longer observed move-time buckets have lower Top1 accuracy and higher NLL. The difference is already visible across the intermediate buckets, rather than appearing only in the small tail of unusually long moves.

Table 16 reports the corresponding values. The fastest bucket, $[0, 2]$ seconds, contains 350,498 positions and has Top1 accuracy above 69% for both models. Accuracy then declines steadily across the fixed-second buckets. In the $(15, 30]$ second bucket, MAIA3 reaches 33.869% Top1 and Allie reaches 31.934%; in the $(30, \infty)$ bucket, these values fall further to 30.423% and 28.856%. Over the same range, NLL rises from 0.892 to 2.226 for MAIA3 and from 0.924 to 2.336 for Allie.

The fixed-second buckets reproduce the main-text pattern, showing that the association between observed move time and model performance is not an artifact of the primary bucket boundaries. These comparisons remain unadjusted and do not isolate move time from correlated aspects of the position or game context.

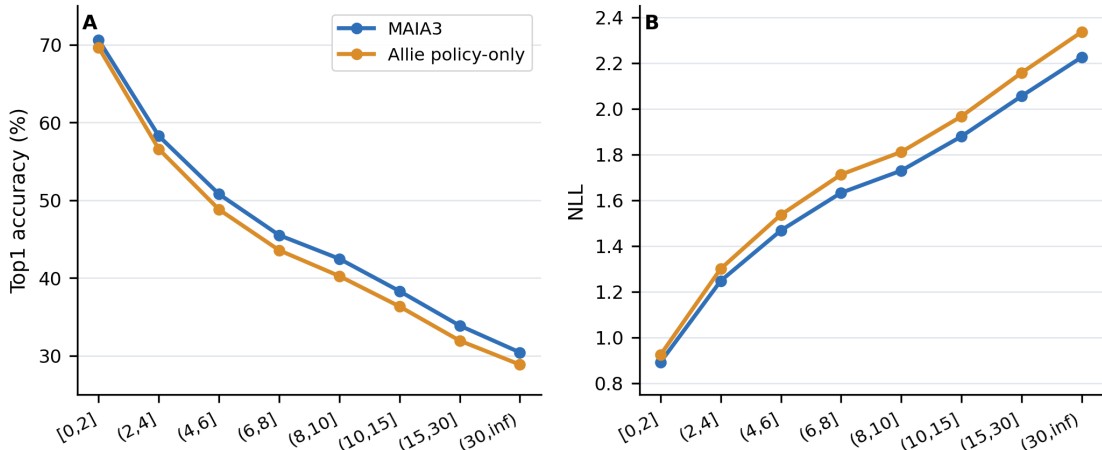

Figure 6: Fixed-second move-time analysis on the shared-protocol Allie blitz evaluation set. Longer observed blitz move times are associated with lower Top1 accuracy and higher NLL for both MAIA3 and Allie policy-only.

Table 16: Fixed-second move-time analysis on the shared-protocol Allie blitz evaluation set. Row counts are exact, and shares are computed over all 884,049 positions.

| Bucket | Rows | % Rows | MAIA3 Top1 | Allie Top1 | MAIA3 NLL | Allie NLL |
|---|---|---|---|---|---|---|
| $[0, 2]$ | 350,498 | 39.65 | 70.612 | 69.633 | 0.892 | 0.924 |
| $(2, 4]$ | 185,249 | 20.95 | 58.257 | 56.578 | 1.247 | 1.300 |
| $(4, 6]$ | 108,127 | 12.23 | 50.783 | 48.805 | 1.468 | 1.536 |
| $(6, 8]$ | 69,746 | 7.89 | 45.501 | 43.559 | 1.632 | 1.712 |
| $(8, 10]$ | 46,410 | 5.25 | 42.467 | 40.243 | 1.729 | 1.811 |
| $(10, 15]$ | 62,813 | 7.11 | 38.304 | 36.335 | 1.879 | 1.967 |
| $(15, 30]$ | 49,718 | 5.62 | 33.869 | 31.934 | 2.055 | 2.157 |
| $(30, \infty)$ | 11,488 | 1.30 | 30.423 | 28.856 | 2.226 | 2.336 |

## C.2 Clock-Context Stratification

Elapsed move time does not fully describe the player's temporal situation. A two-second move may represent little commitment when several minutes remain, but a substantial fraction of the available clock in a time-pressure position. We therefore also stratify performance by the remaining clock before the move, by whether the time control includes an increment, and by the fraction of the pre-move clock spent on the move. If $r_i$ denotes the remaining clock immediately before move $i$, we define the fraction of the available clock spent on the move as

$$\rho_i = \frac{\tau_i}{r_i}.$$

Table 17 reports the resulting strata. Each block separately partitions the full evaluation set, and the percentages therefore sum to 100 within each clock-context variable.

The clearest clock-context pattern is associated with the fraction of clock spent. When a move uses at most 1% of the available clock, MAIA3 reaches 71.674% Top1 and Allie reaches 71.075%. When the move uses more than 15% of the available clock, these values fall to 35.810% and 33.196%, while NLL rises sharply for both models.

The remaining-clock buckets behave differently. Positions with less clock remaining have higher Top1 and lower NLL on average, although the final $= 30$ second bucket contains only 0.21% of the data. This pattern may reflect the relationship between remaining clock, move number, game phase, and the types of positions

Table 17: Prediction quality across observed clock-context strata. The clock fraction is the time spent on the move divided by the remaining clock immediately before the move. Each block partitions all 884,049 positions.

| Bucket | % Rows | MAIA3 Top1 | Allie Top1 | MAIA3 NLL | Allie NLL |
|---|---|---|---|---|---|
| *Remaining clock before the move* | | | | | |
| > 180 s | 29.16 | 54.654 | 53.758 | 1.394 | 1.424 |
| (120, 180] s | 39.15 | 56.960 | 55.740 | 1.300 | 1.345 |
| (60, 120] s | 23.60 | 59.742 | 57.275 | 1.201 | 1.294 |
| (30, 60] s | 7.89 | 60.777 | 58.333 | 1.155 | 1.251 |
| = 30 s | 0.21 | 61.551 | 58.251 | 1.140 | 1.212 |
| *Fraction of the pre-move clock spent* | | | | | |
| ≤ 1% | 21.14 | 71.674 | 71.075 | 0.873 | 0.889 |
| (1, 3]% | 36.42 | 61.704 | 60.508 | 1.153 | 1.193 |
| (3, 7]% | 25.05 | 50.438 | 48.347 | 1.483 | 1.556 |
| (7, 15]% | 12.37 | 42.018 | 39.561 | 1.751 | 1.855 |
| > 15% | 5.02 | 35.810 | 33.196 | 1.984 | 2.122 |
| *Increment* | | | | | |
| No increment | 73.82 | 57.404 | 55.830 | 1.286 | 1.344 |
| Increment > 0 | 26.18 | 56.836 | 55.462 | 1.311 | 1.361 |

reached later in a game. The presence of an increment is associated with only a modest difference in performance.

The clock-context results complement the raw move-time analysis. Moves that consume a larger fraction of the available clock are associated with lower Top1 and higher NLL. These relationships remain descriptive: move time, remaining clock, game phase, legal-move count, rating, model uncertainty, and position difficulty are not varied independently in this benchmark. The analyses therefore characterize performance across temporal groups but do not identify a causal effect of move time or clock context.

# D    Additional Robustness Analyses

The preceding analyses examine the argmax gap through rank geometry, model complementarity, and observed move time. We now consider two further checks. First, we vary how the MAIA3 and Allie distributions are combined, rather than selecting one model's top move. Second, we examine how the Top5–Top1 discrepancy varies across game phase, player rating, branching factor, and model uncertainty.

## D.1    Probability-Ensemble Weight Sweep

The main text reports convex and geometric ensembles with a MAIA3 weight of $\alpha = 0.70$, selected by NLL on held-out data. Here we show how NLL and Top1 vary across the complete mixture-weight sweep.

Let $\pi_i^M$ and $\pi_i^A$ denote the MAIA3 and Allie legal-move distributions on position $i$. The convex mixture is

$$\pi_{i,\alpha}^{\text{conv}}(a) = \alpha \pi_i^M(a) + (1 - \alpha)\pi_i^A(a),$$

whereas the geometric mixture is

$$\pi_{i,\alpha}^{\text{geom}}(a) = \frac{\left(\pi_i^M(a)\right)^\alpha \left(\pi_i^A(a)\right)^{1-\alpha}}{\sum_{b \in A_i} \left(\pi_i^M(b)\right)^\alpha \left(\pi_i^A(b)\right)^{1-\alpha}}.$$

Thus, $\alpha = 1$ recovers MAIA3 and $\alpha = 0$ recovers Allie policy-only.

Figure 7 reports the paper-test curves. Both mixture types improve NLL over a broad range of MAIA3-heavy weights. Top1 behaves differently: it changes only slightly near the MAIA3 endpoint and remains far below the diagnostic oracle throughout the sweep.

Table 18 gives the held-out-selected results. The convex mixture improves NLL from 1.292 to 1.289 and produces 13,468 rescues against 13,323 breaks relative to MAIA3. The resulting net gain of 145 positions

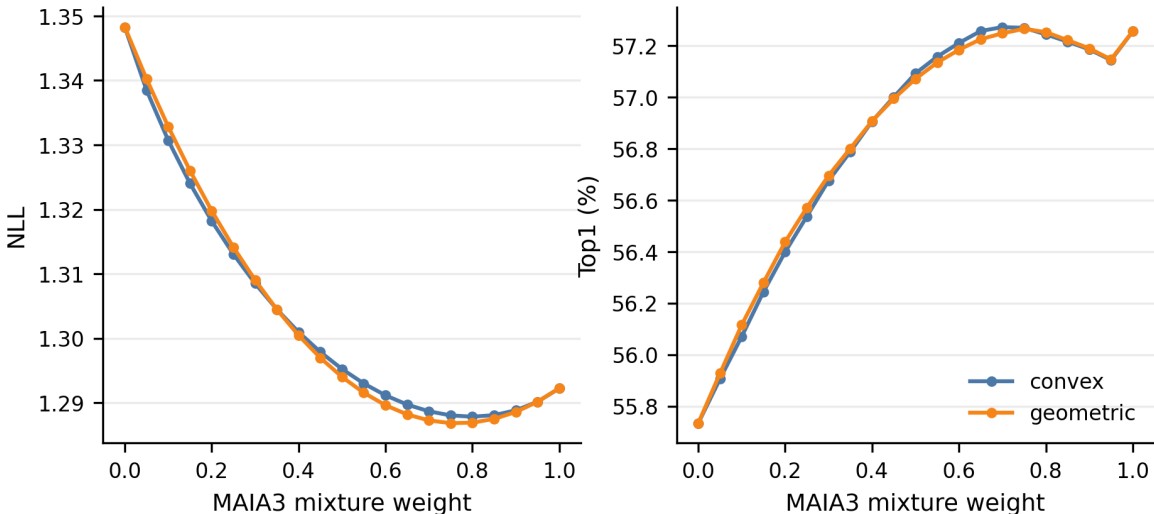

Figure 7: Probability-ensemble sweep over the MAIA3 mixture weight. The left panel reports NLL and the right panel reports Top1 accuracy for convex and geometric mixtures. The main-text weight, $\alpha = 0.70$, is selected by NLL on held-out data rather than from the paper-test sweep.

Table 18: Held-out-selected probability ensembles on the shared-protocol Allie blitz evaluation set. Rescue and break counts are measured relative to MAIA3.

| Method | $\alpha$ | NLL | Top1 | Top5 | Rescues | Breaks | Net |
|---|---|---|---|---|---|---|---|
| MAIA3 | 1.00 | 1.292239 | 57.2549 | 91.8431 | – | – | – |
| Allie policy-only | 0.00 | 1.348264 | 55.7339 | 90.9324 | 40,146 | 53,592 | -13,446 |
| Convex ensemble | 0.70 | 1.288666 | 57.2713 | 91.8368 | 13,468 | 13,323 | +145 |
| Geometric ensemble | 0.70 | 1.287252 | 57.2473 | 91.8516 | 14,404 | 14,471 | -67 |

corresponds to only 0.016 percentage points in Top1. The geometric mixture improves NLL slightly further, to 1.287, but produces 67 more breaks than rescues and finishes slightly below MAIA3 in Top1.

The sweep shows that the ensemble finding is not tied to one isolated weight. Probability averaging improves distributional fit across a range of mixtures, but its effect on Top1 remains negligible relative to the 4.541 percentage point diagnostic-oracle difference.

## D.2 Chess-Context Stratification

The headline Top5–Top1 values average over positions that differ substantially in game stage, player strength, and number of legal choices. We therefore repeat the analysis across game phase, rating, legal-move count, and model-uncertainty strata. These analyses describe how often the single observed human move is ranked within Top5 rather than Top1; they do not establish that the models' Top5 candidates represent the complete set of human-plausible moves. Engine evaluations are not available for the complete shared evaluation set, so we use legal-move count as a model-independent measure of branching and MAIA3 entropy and margin as model-based measures of uncertainty.

**Game phase and branching factor.** We use a move-number-based phase proxy: moves 1–10, 11–40, and 41 onward define the opening, middlegame, and endgame groups, respectively. Figure 8 summarizes the phase and branching-factor results. The Top5–Top1 discrepancy narrows from the opening to the endgame for both models. A similar pattern appears with legal-move count: positions with 1–20 legal moves have a noticeably smaller discrepancy, whereas the well-populated groups with more than 20 legal moves remain around 35–37 percentage points.

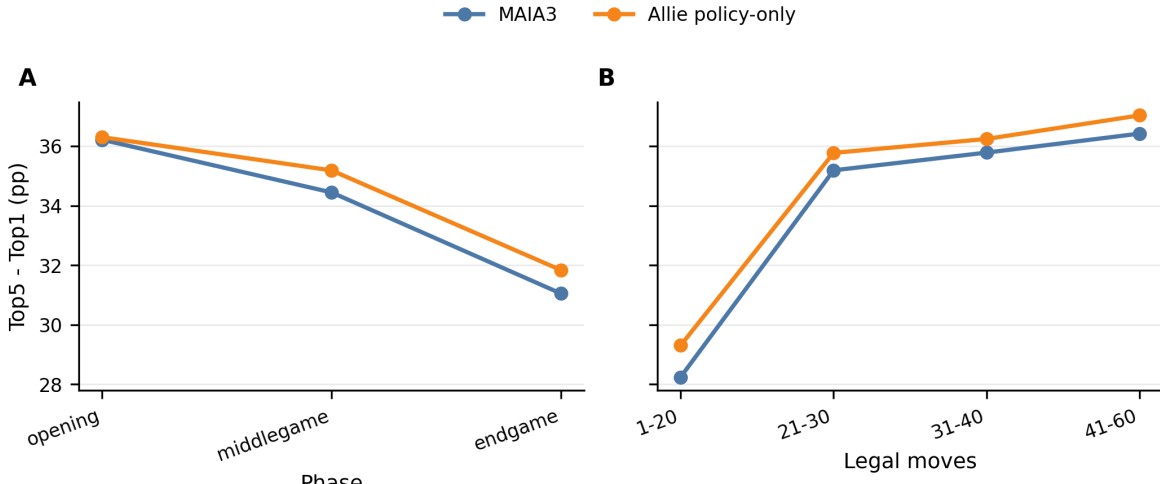

Figure 8: Top5–Top1 discrepancy across game phase and number of legal moves. The legal-move groups are ordered by increasing branching factor. The discrepancy narrows in endgames and lower-branching positions, but remains substantial across the well-populated strata. The group with more than 60 legal moves is omitted from the right panel because it contains only 368 positions; it is retained in Table 20.

Table 19 gives the phase results. MAIA3's Top5–Top1 discrepancy falls from 36.216 percentage points in the opening to 31.049 in the endgame. Allie policy-only follows the same pattern, with corresponding values of 36.303 and 31.836 percentage points.

Table 19: Top1, Top5, and the Top5–Top1 discrepancy by game phase. Performance and discrepancy values are percentages and percentage points, respectively.

| Phase | Rows | M3 Top1 | M3 Top5 | M3 Gap | Allie Top1 | Allie Top5 | Allie Gap |
|---|---|---|---|---|---|---|---|
| Opening | 178,354 | 54.433 | 90.649 | 36.216 | 54.302 | 90.605 | 36.303 |
| Middlegame | 649,661 | 57.383 | 91.829 | 34.447 | 55.501 | 90.686 | 35.185 |
| Endgame | 56,034 | 64.755 | 95.804 | 31.049 | 62.996 | 94.832 | 31.836 |

Table 20 reports the corresponding branching-factor results. Positions with 1–20 legal moves have higher Top1 and smaller Top5–Top1 discrepancies: 28.246 percentage points for MAIA3 and 29.307 for Allie. Once the position contains more than 20 legal moves, the discrepancy remains above 35 percentage points across the well-populated groups.

Table 20: Top1, Top5, and the Top5–Top1 discrepancy by number of legal moves. The final group is included for completeness but contains only 368 positions.

| Legal moves | Rows | M3 Top1 | M3 Top5 | M3 Gap | Allie Top1 | Allie Top5 | Allie Gap |
|---|---|---|---|---|---|---|---|
| 1–20 | 139,380 | 69.297 | 97.543 | 28.246 | 67.807 | 97.114 | 29.307 |
| 21–30 | 196,497 | 56.392 | 91.580 | 35.188 | 54.857 | 90.630 | 35.774 |
| 31–40 | 374,523 | 54.850 | 90.634 | 35.783 | 53.471 | 89.712 | 36.242 |
| 41–60 | 173,281 | 53.770 | 90.191 | 36.421 | 51.930 | 88.963 | 37.033 |
| > 60 | 368 | 45.380 | 82.337 | 36.957 | 45.924 | 79.891 | 33.967 |

**Player rating.** Table 21 shows how the Top5–Top1 discrepancy varies with player rating. Top1 accuracy generally increases with rating, while the MAIA3 discrepancy decreases from roughly 36 percentage points at lower ratings to roughly 33 percentage points across the well-populated higher-rating groups. The final two rating groups contain relatively few positions and should be interpreted cautiously.

Table 21: Top5–Top1 discrepancy and diagnostic-oracle accuracy by player-rating stratum. Discrepancy values are in percentage points and oracle values are percentages.

| Rating | Rows | MAIA3 gap | Allie gap | Oracle Top1 |
|---|---|---|---|---|
| 600–800 | 28,891 | 36.423 | 36.378 | 52.338 |
| 800–1000 | 79,920 | 36.269 | 36.240 | 55.795 |
| 1000–1200 | 91,966 | 35.682 | 35.826 | 57.984 |
| 1200–1400 | 92,848 | 35.583 | 35.630 | 59.358 |
| 1400–1600 | 97,310 | 34.768 | 35.082 | 60.960 |
| 1600–1800 | 105,872 | 34.599 | 35.032 | 61.737 |
| 1800–2000 | 105,472 | 34.278 | 34.873 | 63.143 |
| 2000–2200 | 102,817 | 33.934 | 34.877 | 64.709 |
| 2200–2400 | 105,634 | 32.993 | 34.511 | 66.666 |
| 2400–2600 | 59,968 | 32.804 | 34.575 | 67.985 |
| 2600–2800 | 12,392 | 32.941 | 34.345 | 68.383 |
| 2800–3000 | 878 | 31.435 | 37.244 | 70.729 |
| 3000–3200 | 81 | 30.864 | 37.037 | 71.605 |

**Model uncertainty.** We define uncertainty using MAIA3's legal-move entropy and the probability margin between its first- and second-ranked moves. Both quantities are available without observing the target move.

Table 22 shows that the Top5–Top1 discrepancy is larger in strata where MAIA3 is less confident. In the lowest-entropy quintile, MAIA3 reaches 92.116% Top1 and has a discrepancy of only 7.387 percentage points. As entropy increases, Top1 falls much faster than Top5, producing discrepancies above 40 percentage points in the upper three quintiles.

The margin analysis gives the same qualitative picture. When the MAIA3 Top1–Top2 margin exceeds 0.25, its discrepancy is 22.713 percentage points. When the margin is at most 0.10, the discrepancy exceeds 53 percentage points. The discrepancy is therefore largest when the predicted distribution does not strongly separate its leading candidates.

Table 22: Top5–Top1 geometry across MAIA3 uncertainty strata. Entropy groups are equal-frequency quintiles. Margin groups are defined using the MAIA3 Top1–Top2 probability difference.

| Scheme | Stratum | Rows | M3 Top1 | M3 Top5 | M3 Gap | Allie Gap |
|---|---|---|---|---|---|---|
| Entropy | Q1, lowest | 176,810 | 92.116 | 99.503 | 7.387 | 7.901 |
| Entropy | Q2 | 176,810 | 68.159 | 98.114 | 29.955 | 31.427 |
| Entropy | Q3 | 176,809 | 54.050 | 95.150 | 41.100 | 42.267 |
| Entropy | Q4 | 176,810 | 42.100 | 89.674 | 47.574 | 48.011 |
| Entropy | Q5, highest | 176,810 | 29.850 | 76.775 | 46.925 | 46.387 |
| Margin | $\leq 0.03$ | 64,312 | 30.557 | 84.065 | 53.508 | 53.202 |
| Margin | $(0.03, 0.10]$ | 124,426 | 32.525 | 85.665 | 53.140 | 52.553 |
| Margin | $(0.10, 0.25]$ | 190,937 | 41.164 | 88.661 | 47.497 | 48.594 |
| Margin | $> 0.25$ | 504,374 | 72.851 | 95.564 | 22.713 | 23.551 |

The stratified analyses refine the headline result without changing it. The magnitude of the Top5–Top1 discrepancy varies across game phase, player rating, branching factor, and model uncertainty. It is smaller in endgames, lower-branching positions, and more confident predictions, but remains substantial across the well-populated strata. These descriptive associations do not determine why the observed move is not ranked first or how much of the discrepancy is reducible through better modeling.

# E   Statistical Validation

All intervention methods are evaluated on the same 884,049 positions, so their Top1 differences can be assessed row by row. For each comparison, we report the change in Top1, a paired position-level 95% confidence interval, and a two-sided McNemar test based on the rescue and break counts. Because the evaluation set is large, small effects can produce very small *p*-values; we therefore interpret the results

primarily through the effect sizes, confidence intervals, and rescue/break balance rather than statistical significance alone. Unless explicitly labeled as game-clustered, the reported intervals treat positions as the evaluation unit and do not adjust for dependence among positions originating from the same game. For the additional nonlinear selector and MRR/NDCG analyses, we also report 10k game-clustered bootstrap intervals as a robustness check. For the conservative rank-2 correction gate, we likewise report a 10k game-clustered interval.

### E.1 Paired Top1 Comparisons

Table 23 reports the paired comparisons underlying the main Top1 results. The diagnostic oracle has a large positive difference relative to MAIA3, showing that the two models make complementary correct predictions. Because it uses the observed human move label, this oracle is a descriptive upper bound rather than a deployable intervention. In contrast, the position-level confidence intervals for the fixed selectors and the pre-move linear cross-model trained selector lie below zero. The MAIA3 shortlist selector is the closest case: its point estimate is slightly negative, but its interval includes zero.

The pre-move linear Allie shortlist selector improves Allie policy-only by 1.432 percentage points, with a position-level interval that lies entirely above zero. This shows that the tested reranking method improves the weaker base policy without using realized move duration. When compared directly with MAIA3, however, its Top1 remains 0.089 percentage points lower. The conservative rank-2 correction gate improves MAIA3 by 0.137 percentage points. Its position-level interval is entirely above zero, and its 10k game-clustered interval is also $[+0.100, +0.173]$ percentage points. The probability ensembles produce much smaller changes: the convex ensemble has a point gain of 0.016 percentage points and the geometric ensemble a point loss of 0.008 percentage points, with both position-level intervals including zero.

Table 23: Position-level paired Top1 comparisons on the shared-protocol Allie blitz evaluation set. Deltas and confidence intervals are in percentage points. Rescues and breaks are measured relative to the base model in each comparison. The learned-selector rows use the 45-feature pre-move specification.

| Comparison | $\Delta$ Top1 | 95% CI | Rescues | Breaks | $p$-value |
|---|---|---|---|---|---|
| Allie policy-only vs. MAIA3 | -1.521 | $[-1.589, -1.454]$ | 40,146 | 53,592 | $< 10^{-300}$ |
| MAIA3–Allie oracle vs. MAIA3 | +4.541 | $[4.496, 4.584]$ | 40,146 | 0 | $< 10^{-300}$ |
| raw max-p selector vs. MAIA3 | -0.216 | $[-0.263, -0.171]$ | 19,962 | 21,874 | $9.37 \times 10^{-21}$ |
| min-entropy selector vs. MAIA3 | -0.292 | $[-0.332, -0.248]$ | 15,954 | 18,534 | $7.56 \times 10^{-44}$ |
| max-margin selector vs. MAIA3 | -0.245 | $[-0.289, -0.196]$ | 21,171 | 23,333 | $1.26 \times 10^{-24}$ |
| cross-model trained selector vs. MAIA3 | -0.203 | $[-0.248, -0.155]$ | 21,596 | 23,392 | $2.61 \times 10^{-17}$ |
| MAIA3 shortlist selector vs. MAIA3 | -0.018 | $[-0.056, 0.017]$ | 12,909 | 13,067 | 0.330 |
| Allie shortlist selector vs. Allie | +1.432 | $[1.380, 1.484]$ | 33,004 | 20,348 | $< 10^{-300}$ |
| Allie shortlist selector vs. MAIA3 | -0.089 | $[-0.136, -0.044]$ | 21,044 | 21,834 | $1.39 \times 10^{-4}$ |
| Conservative rank-2 gate vs. MAIA3 | +0.137 | $[+0.100, +0.173]$ | 14,176 | 12,969 | $2.48 \times 10^{-13}$ |
| Convex ensemble vs. MAIA3 | +0.016 | $[-0.021, 0.055]$ | 13,468 | 13,323 | 0.379 |
| Geometric ensemble vs. MAIA3 | -0.008 | $[-0.046, 0.031]$ | 14,404 | 14,471 | 0.698 |
| MAIA3 trust-region refiner vs. MAIA3 | -0.167 | $[-0.176, -0.158]$ | 0 | 1,474 | $< 10^{-300}$ |
| Allie trust-region refiner vs. Allie | +0.000 | $[0.000, 0.000]$ | 0 | 0 | 1 |

The paired comparisons agree with the rescue/break accounting in the main text. The oracle summarizes complementary correct predictions, but the fixed and primary learned MAIA3-relative selectors do not produce a positive Top1 change. For the fixed and cross-model trained selectors, breaks outnumber rescues. The MAIA3 shortlist selector nearly balances the two, and its observed change is not distinguishable from zero under the position-level paired analysis.

The Allie shortlist selector presents a different case. Its rescues substantially exceed its breaks relative to Allie, producing a positive Top1 change over that base policy. Nevertheless, its final accuracy remains slightly below MAIA3. The conservative rank-2 gate also has more rescues than breaks, producing a small positive gain over MAIA3. It rescues 14,176 MAIA3 errors and breaks 12,969 MAIA3-correct predictions, for a net gain of 1,207 positions. The convex ensemble also has slightly more rescues than breaks, but the difference is only 145 positions and its position-level interval includes zero. Its small positive point estimate should therefore not be interpreted as meaningful recovery of the diagnostic-oracle difference.

Table 24 reports statistical validation for the nonlinear selector baselines. These baselines use the same audited pre-move feature matrix as the primary learned selectors and exclude realized target-move duration. The cross-model MLP and XGBoost selectors remain below MAIA3, as does the MAIA3 shortlist XGBoost selector. The Allie shortlist XGBoost selector improves Allie but remains below MAIA3. Thus, the nonlinear baselines do not change the selector conclusion.

Table 24: Paired Top1 comparisons for nonlinear selector baselines. Deltas and confidence intervals are in percentage points. Game-clustered intervals use 10k bootstrap samples over source games.

| Comparison | $\Delta$ Top1 | Position CI | Game CI | Rescues | Breaks | $p$-value |
|---|---|---|---|---|---|---|
| Cross-model MLP selector vs. MAIA3 | -0.172 | $[-0.215, -0.131]$ | $[-0.215, -0.130]$ | 16,835 | 18,358 | $4.93\times10^{-16}$ |
| Cross-model XGBoost selector vs. MAIA3 | -0.158 | $[-0.170, -0.146]$ | $[-0.170, -0.146]$ | 743 | 2,140 | $5.03\times10^{-149}$ |
| MAIA3 shortlist XGBoost selector vs. MAIA3 | -0.165 | $[-0.173, -0.156]$ | $[-0.173, -0.156]$ | 77 | 1,532 | $1.06\times10^{-287}$ |
| Allie shortlist XGBoost selector vs. Allie | +0.688 | $[+0.660, +0.716]$ | $[+0.659, +0.718]$ | 10,914 | 4,828 | $< 10^{-300}$ |

### E.2 Probability-Level Comparisons

For NLL, we compare the per-position losses produced by each intervention and its corresponding base model. Table 25 reports the resulting position-level differences and 95% confidence intervals. Negative values indicate that the intervention assigns greater probability to the observed human move on average.

All reported NLL intervals lie below zero. Calibration improves NLL for both base models while preserving their rankings and therefore leaving Top1 unchanged. Trust-region refinement also improves NLL, but reduces MAIA3 Top1 and leaves Allie Top1 unchanged. The convex and geometric ensembles likewise improve NLL relative to MAIA3, while their Top1 changes remain close to zero.

Table 25: Position-level paired NLL comparisons for calibration, trust-region refinement, and probability ensembling. Negative $\Delta$ NLL indicates improved likelihood.

| Comparison | $\Delta$ NLL | 95% CI | $\Delta$ Top1 (pp) |
|---|---|---|---|
| MAIA3 calibrated vs. MAIA3 base | -0.0081 | $[-0.0084, -0.0079]$ | +0.000 |
| Allie calibrated vs. Allie base | -0.0101 | $[-0.0104, -0.0098]$ | +0.000 |
| Convex ensemble vs. MAIA3 | -0.0036 | $[-0.0039, -0.0033]$ | +0.016 |
| Geometric ensemble vs. MAIA3 | -0.0050 | $[-0.0053, -0.0047]$ | -0.008 |
| MAIA3 trust-region refiner vs. MAIA3 base | -0.0072 | $[-0.0074, -0.0070]$ | -0.167 |
| Allie trust-region refiner vs. Allie base | -0.0087 | $[-0.0089, -0.0084]$ | +0.000 |

These comparisons make the NLL–Top1 divergence explicit. The tested probability-level interventions improve likelihood, but their corresponding Top1 changes are zero, negative, or too small to distinguish from zero under the position-level analysis. Assigning greater probability to the observed human move therefore does not necessarily make the induced Top1 prediction more accurate.

Table 26 reports paired MRR and NDCG@5 deltas for the probability-level methods. These metrics provide a midpoint between NLL and Top1 by asking whether the observed human move moves closer to the top of the ranked list.

Table 26: Paired MRR and NDCG@5 comparisons for probability-level methods. Confidence intervals are raw metric units. Game-clustered intervals use 10k bootstrap samples over source games.

| Comparison | $\Delta$ MRR | MRR game CI | $\Delta$ NDCG@5 | NDCG@5 game CI |
|---|---|---|---|---|
| MAIA3 calibrated vs. base | +0.000000 | $[+0.000000, +0.000000]$ | +0.000000 | $[+0.000000, +0.000000]$ |
| MAIA3 trust-region vs. base | -0.001162 | $[-0.001207, -0.001117]$ | -0.001068 | $[-0.001124, -0.001013]$ |
| Allie calibrated vs. base | +0.000000 | $[+0.000000, +0.000000]$ | +0.000000 | $[+0.000000, +0.000000]$ |
| Allie trust-region vs. base | -0.000001 | $[-0.000006, +0.000004]$ | -0.000007 | $[-0.000057, +0.000043]$ |
| Convex ensemble vs. MAIA3 | +0.000106 | $[-0.000091, +0.000308]$ | +0.000057 | $[-0.000122, +0.000237]$ |
| Geometric ensemble vs. MAIA3 | +0.000006 | $[-0.000206, +0.000215]$ | +0.000029 | $[-0.000159, +0.000212]$ |

Calibration leaves MRR and NDCG exactly unchanged because it is rank-preserving. MAIA3 trust-region refinement improves NLL but worsens both MRR and NDCG@5. Convex and geometric ensembles have tiny positive point changes in MRR and NDCG@5, but their game-clustered confidence intervals include zero. These midpoint ranking metrics therefore support the main conclusion: probability-level improvements do not reliably translate into either hard Top1 recovery or substantial intermediate-rank gains.

Table 27 gives rank-change counts for the same probability-level comparisons. Calibration produces no rank changes. MAIA3 trust-region refinement worsens more ranks than it improves, while the ensemble methods change many ranks in both directions but have nearly balanced Top1 and Top5 entry and exit counts.

Table 27: Rank-change counts for probability-level methods. "Top1 in" and "Top1 out" count rows moving into or out of exact Top1 correctness; "Top5 in" and "Top5 out" are defined analogously.

| Comparison | Improved | Worsened | Unchanged | Top1 in | Top1 out | Top5 in | Top5 out |
|---|---|---|---|---|---|---|---|
| MAIA3 calibrated vs. base | 0 | 0 | 884,049 | 0 | 0 | 0 | 0 |
| MAIA3 trust-region vs. base | 1,178 | 5,483 | 877,388 | 0 | 1,474 | 1,127 | 1,592 |
| Allie calibrated vs. base | 0 | 0 | 884,049 | 0 | 0 | 0 | 0 |
| Allie trust-region vs. base | 1,709 | 1,691 | 880,649 | 0 | 0 | 1,633 | 1,647 |
| Convex ensemble vs. MAIA3 | 53,904 | 57,867 | 772,278 | 13,468 | 13,323 | 5,429 | 5,485 |
| Geometric ensemble vs. MAIA3 | 58,298 | 55,848 | 769,903 | 14,404 | 14,471 | 5,494 | 5,419 |

### E.3 Move-Time Differences

The move-time buckets contain different positions, so their comparisons are not paired intervention tests. We report position-level intervals for the difference between the fastest and slowest primary buckets. For Top1, the intervals use a normal approximation for two binomial proportions. For NLL, they use a Welch normal approximation (Welch, 1947a;b) over the per-position losses in the two disjoint buckets. These calculations do not adjust for positions being grouped within games.

Table 28: Position-level differences between the slowest and fastest observed blitz move-time buckets. The fastest bucket is $[0, 1]$ seconds and the slowest is $(7, \infty)$ seconds. Top1 differences are in percentage points.

| Model | Metric | Fast | Slow | Slow–fast | 95% CI |
|---|---|---|---|---|---|
| MAIA3 | Top1 | 73.405 | 38.625 | -34.780 | $[-35.073, -34.487]$ |
| Allie policy-only | Top1 | 72.596 | 36.638 | -35.958 | $[-36.250, -35.665]$ |
| MAIA3 | NLL | 0.812 | 1.876 | +1.064 | $[1.056, 1.072]$ |
| Allie policy-only | NLL | 0.837 | 1.967 | +1.130 | $[1.122, 1.138]$ |

The observed differences between the fastest and slowest groups are large in magnitude: for both models, the slowest group has roughly 35 percentage points lower Top1 and more than one additional NLL unit. These comparisons describe an unadjusted association between observed move time and model performance. They do not identify a causal effect of move time or isolate it from correlated factors such as game phase, legal-move count, rating, model uncertainty, and clock state.

Overall, the position-level analyses are consistent with the paper's descriptive findings. MAIA3 and Allie make complementary correct predictions, but the fixed selectors, primary linear learned selectors, small-MLP and XGBoost learned selector baselines, and probability ensembles do not produce a meaningful Top1 improvement over MAIA3. The conservative rank-2 correction gate is an exception: it produces a small but statistically reliable Top1 gain while recovering only a small fraction of the diagnostic-oracle headroom. Calibration, refinement, and ensembling improve likelihood without producing corresponding Top1 gains. The added MRR/NDCG analyses show that these probability-level gains also do not produce substantial intermediate-rank improvements. Model performance also differs substantially across observed move-time groups, although these comparisons involve different positions and correlated game contexts.

# F   Data Provenance and Split Independence

This appendix documents the construction of the shared-protocol evaluation set, the base-model checkpoints, the separation between training, development, and final evaluation data, and the legal-move processing used to compare MAIA3 and Allie policy-only.

## F.1   Evaluation-Set Construction

The source data are the Allie Lichess 2022 blitz test split (Zhang et al., 2024), containing 1,239,353 half-move positions from 18,239 games. We replay each game using `python-chess` version 1.11.2 and reconstruct the pre-move board, legal UCI moves, observed human move, ratings, clocks, and time-control fields.

Two protocol restrictions determine which target positions are retained. First, target plies 1–10, corresponding to the first five full moves, are not considered; ply 11 is the first eligible target. These initial plies remain available as preceding game history. Second, before emitting a target row, the converter checks the side-to-move player's remaining clock immediately before the move. If this value is strictly below 30 seconds, the current target and all later targets from that game are omitted. A target with exactly 30 seconds remaining is retained.

The 30-second restriction applies to the remaining clock before the move, not to the time spent playing the move. A retained move may therefore take more than 30 seconds, provided that the player has at least 30 seconds remaining before the move begins.

After these restrictions are applied, we verify that the observed move aligns with the reconstructed pre-move position, the legal-move set is recoverable, the required context fields are available, and both model outputs are present. These validation checks remove no additional positions. The resulting shared-protocol evaluation set contains 884,049 positions from 18,138 games, corresponding to 71.33% of the source positions.

Table 29:   Sequential construction of the shared-protocol Allie blitz evaluation set.

| Stage | Positions | Games |
|---|---|---|
| Source Allie blitz test split | 1,239,353 | 18,239 |
| After excluding target plies 1–10 | 1,057,010 | 18,146 |
| After pre-move clock truncation | 884,049 | 18,138 |
| After alignment and output validation | 884,049 | 18,138 |

The first restriction excludes 182,343 potential target positions, while the remaining-clock restriction excludes a further 172,961 positions. The conversion found no move/time-length mismatches, invalid time controls, target-move mismatches, or unrecoverable legal-move sets.

## F.2   Filtering Sensitivity

The retained subset differs from the source split because the protocol removes early target positions and truncates the low-clock tail of each game. Table 30 reports the largest continuous and phase-related changes.

No continuous characteristic has an absolute SMD above 0.20. The largest continuous shifts are in legal-move count and target-move time. Player and opponent ratings change only slightly, with SMDs of approximately 0.024, and time-control composition is also comparatively stable. Phase composition changes more visibly because early targets and low-clock game tails are excluded.

Cached Allie predictions are unavailable for the excluded source positions. We therefore do not compare model performance before and after filtering, since doing so would require additional model inference. All reported results apply specifically to the shared-protocol evaluation set.

Table 30: Largest shifts between the source split and the retained evaluation set. Continuous changes are reported as standardized mean differences (SMD), while phase changes are percentage-point differences.

| Characteristic | Source | Retained | Change |
|---|---|---|---|
| Mean legal moves | 29.275 | 31.206 | SMD +0.163 |
| Mean target-move time (s) | 4.731 | 5.623 | SMD +0.130 |
| Mean pre-move clock (s) | 150.603 | 157.549 | SMD +0.090 |
| Mean move number | 21.452 | 20.741 | SMD −0.052 |
| Opening share | 29.11% | 20.17% | −8.94 pp |
| Middlegame share | 58.38% | 73.49% | +15.10 pp |
| Endgame share | 12.50% | 6.34% | −6.17 pp |

### F.3 Model and Checkpoint Provenance

For MAIA3, we use the public 79M policy checkpoint `maia3-79m.pt` from the `UofTCSSLab/Maia3-79M` release.

For Allie policy-only, we use the public medium policy checkpoint `best.pt` with the medium model configuration supplied by the Allie release.

These identifiers specify the model files used to generate the cached policy outputs. The relationship between the original model-training data and the evaluation set is documented in Appendix F.6.

### F.4 Development and Evaluation Split Independence

The learned selectors, calibration parameters, trust-region configurations, and ensemble weights are fitted or selected using a separate 2023 held-out development artifact. Some procedures draw from the same held-out source; in particular, calibration fitting and ensemble-weight selection use the same 500,000-position sample. These development procedures should therefore not be interpreted as using mutually independent datasets.

The relevant requirement for final evaluation is that no fitting or selection split shares evaluation rows or source games with the shared-protocol evaluation set. Table 31 confirms zero row-key and zero game-ID overlap in every comparison.

Table 31: Overlap between the shared-protocol evaluation set and each downstream fitting or selection split. Canonical-position overlap records repeated board states across distinct games; context-position overlap additionally accounts for the preceding game history.

| Split | Rows | Row overlap | Game overlap | Canonical overlap | Context overlap |
|---|---|---|---|---|---|
| Selector training | 400,000 | 0 | 0 | 7,755 | 0 |
| Selector validation | 100,000 | 0 | 0 | 2,759 | 0 |
| Calibration fitting | 500,000 | 0 | 0 | 8,989 | 0 |
| Ensemble selection | 500,000 | 0 | 0 | 8,989 | 0 |
| Refiner training | 400,000 | 0 | 0 | 7,832 | 0 |
| Refiner validation | 100,000 | 0 | 0 | 2,648 | 0 |

Canonical board positions are defined using the first four FEN fields: piece placement, side to move, castling rights, and en-passant target. Such positions can recur independently across games and years and are therefore reported rather than removed. The context-sensitive key additionally distinguishes the preceding game history and has zero overlap in every comparison.

The evaluation set contains 870,517 unique canonical positions. A total of 4,798 canonical positions are associated with more than one observed human move. These repetitions are retained because they correspond to distinct human decisions and can differ in move history, player context, clock state, or the move played. They are not treated as repeated observations of an identical prediction context or as direct estimates of the conditional human-move distribution.

### F.5 Legal-Move Alignment and Probability Processing

Both policies are evaluated over the same reconstructed legal-move set. MAIA3 reconstructs the board from the stored pre-move FEN, validates the legal UCI moves, masks nonlegal vocabulary entries, and applies log-softmax over the legal entries. Allie reconstructs the target pre-move board from the game prefix, gathers the logits corresponding to the legal UCI moves, and applies softmax over those logits.

For either model, if $z_i(a)$ denotes the score assigned to legal move $a$, the normalized probability is

$$p_i(a) = \frac{\exp z_i(a)}{\sum_{b \in A_i} \exp z_i(b)}.$$

Nonlegal moves receive no probability mass.

MAIA3 NLL is computed directly from the legal-move log-softmax. Allie uses a numerical floor of $10^{-45}$ only inside the logarithm used to compute NLL. No evaluation row has zero observed-human-move probability under either policy.

Across all 884,049 evaluation positions, the audit finds no target-move mismatches, missing outputs, unrecoverable legal-move sets, nonfinite probability rows, or duplicate legal-move entries.

Table 32: Summary of legal-move probability validation.

| Model | Valid rows | Maximum probability-sum error |
|---|---|---|
| MAIA3 | 884,049 | $3.55 \times 10^{-7}$ |
| Allie policy-only | 884,049 | $3.80 \times 10^{-7}$ |

Both cached prediction tables pass one-to-one join validation. Top-$k$ correctness uses the strict-greater tie-aware rank defined in Section 2: the observed move's rank is one plus the number of legal moves receiving a strictly greater score.

For clarity, the move-time variable used in the post-hoc analyses is the realized duration of the target move. Neither base policy nor the primary learned selectors receives this target-move duration. The Allie game prefix may contain the durations of earlier moves, because those moves occur before the target decision.

### F.6 Base-Model Training and Evaluation Independence

The training and evaluation games are independent for both base policies.

For Allie, the authors construct a downsampled dataset of Lichess blitz games played in 2022, reserve approximately 18,000 games for testing, and use the remaining games for training and validation (Zhang et al., 2024). The shared-protocol evaluation set used in this paper is derived from that held-out Allie test split. The Allie training and validation games are therefore separate from the evaluation games.

MAIA3 is trained on Lichess blitz games played from January 2023 through July 2025, while the shared-protocol evaluation set is derived from Allie's 2022 test games (Monroe et al., 2026). The MAIA3 training games and the evaluation games are therefore temporally disjoint.

Accordingly, neither base policy is trained on the evaluation games used in this paper. Canonical board positions may naturally recur across different chess games, but such recurrence does not constitute overlap between the training and evaluation game sets.

