# OpenReview forum: "The Argmax Gap in Human Chess Move Prediction"
_TMLR — Under review for TMLR_

### Review · Reviewer_2B2f · 2026-06-30

**Summary Of Contributions:**

This paper highlights the “argmax gap” in predicting human chess moves: powerful models often rank moves actually played by humans high among their candidate moves but fail to list them as the top choice. Across 884,049 rapid chess positions, MAIA3 and Allie achieved Top-5 accuracy rates exceeding 90%, but their Top-1 accuracy rates were only around 56% to 57%. Although these two models provide complementary correct predictions, methods such as confidence-based selectors, learning reorderers, calibration, and probability sets have largely failed to restore the achievable “oracle” improvement, as corrected errors are offset by newly introduced errors. The paper also demonstrates that the longer humans take to think, the less predictable their moves become; furthermore, increasing likelihood or calibration does not necessarily improve the accuracy of Top-1 matches. Overall, the paper argues that predicting human moves should be evaluated comprehensively as a problem involving ranking, probability prediction, and decision selection. The paper’s greatest limitation is that, there is only discussion regarding how the combination of two model doesn't work and the analyze of the reason, but no actual solution that improve the argmax gap.

**Audience:**

Yes

**Audience Explanation:**

The main finding—that strong human-move predictors achieve over 90% Top-5 accuracy but only about 56–57% Top-1 accuracy—highlights a practically meaningful distinction between identifying plausible human actions and selecting the exact action taken. Besides, the paper reveals that improving NLL, calibration, or probability ensembling does not necessarily improve top-1 accuracy.

Since the experiment was limited to rapid chess and two human-play prediction systems, the target audience is likely to be relatively specialized. However, the methodological insights from this study have broader applicability in the fields of behavior prediction and probabilistic classification. Therefore, although this study is unlikely to attract widespread attention, it remains relevant to a portion of the TMLR audience.

**Broader Impact Concerns:**

1. When the classification algorithm is applied to education scenario for chess, there is possibility to collect user data for profiling.
2. The classification algorithm can be treated as a reference for plagiarism, but not as concrete evidence.

**Claims And Evidence:**

Yes

**Claims Explanation:**

Overall, the assessment is positive. The paper’s main arguments are supported by clear and convincing empirical evidence, though certain broader interpretations should still be treated with caution.

The core claim—that there is a significant “argmax gap” between Top 5 and Top 1—is strongly supported. Both MAIA3 and Allie were evaluated on the same dataset of 884,049 chess positions and valid moves, and both demonstrated Top-5 accuracy exceeding 90%, while Top-1 accuracy was only around 56–57%. The paper further confirms that even in positions where the Top-1 prediction was incorrect, the observed moves typically ranked among the top choices.

The argument that the models contain complementary information was also convincingly demonstrated through pairwise correctness analysis: an “oracle” model that selects either correct model achieves a Top-1 accuracy of 61.796%, far exceeding the performance of either model alone. Analysis of the rescue/breakthrough mechanism clearly explains why practical selectors fail to capture this “oracle” benefit.

The evidence regarding the divergence between probability quality and precise prediction is equally compelling. While calibration, refinement, and ensemble methods continuously improve NLL, their impact on Top-1 is negligible, zero, or negative. The credibility of these results is further enhanced by fitting and selection using held-out data, pairwise comparisons, confidence interval analysis, and the McNemar test.

**Requested Changes:**

1. Clarify the novelty: The Top-k Top-1 concept is not a new concept in classification, the paper just applied it to the chess task.
2. Investigate more on how to improve the argmax gap problem, instead of just showing the existence of the problem.
3. Since each board position corresponds to only one observed human move, the paper cannot determine whether the other Top 5 candidates are also reasonable human choices. Therefore, a Top 1 mismatch does not necessarily indicate that the model has made a correctable ranking error; it may also stem from the inherent diversity of human decision-making or unobserved context. There should be more discussion regarding this.

---

> ### Author Response · Authors · 2026-07-16
> **Response to Reviewer 2B2f**
>
> We thank the reviewer for the positive assessment and for the helpful requested changes. We revised the manuscript to clarify the novelty, added a direct improvement attempt, expanded the single-observed-move limitation discussion, and added a Broader Impact section in the updated manuscript.
>
> **Clarifying the novelty**
>
> We agree that Top-k accuracy and Top1 accuracy are standard ranking metrics, and that the Top-k–Top1 difference is not a new classification concept. We revised the Abstract, Introduction, and Problem Formulation to make the framing more precise. We now describe the “argmax gap” as a descriptive label for the empirical Top5–Top1 discrepancy observed in this human-chess-prediction setting, rather than as a new metric.
>
> The contribution is therefore not the definition of Top-k or Top1, but the empirical characterization of how this discrepancy behaves for strong current human-move predictors, how it relates to NLL/calibration/ranking, and why it is difficult to exploit through selection without introducing breaks.
>
> **Adding an improvement attempt**
>
> We agree that the original version was mostly diagnostic. To address this, we added a conservative break-aware correction gate. The method keeps MAIA3’s Top1 prediction by default and switches only to MAIA3’s second-ranked move when a validation-selected pre-move score predicts a positive net gain. Model fitting and threshold selection are done on off-test/validation data, and paper-test labels are used only after selection.
>
> | Method                   |   Top1 | Delta vs. MAIA3 | Rescues | Breaks |    Net | Switch rate |
> | ------------------------ | -----: | --------------: | ------: | -----: | -----: | ----------: |
> | Conservative rank-2 gate | 57.391 |       +0.137 pp |  14,176 | 12,969 | +1,207 |      5.338% |
>
> The gain is small but reliable: both the position-level and 10k game-clustered 95% intervals are [+0.100, +0.173] percentage points, with McNemar p = 2.48e-13. This recovers 3.01% of the 4.541 percentage-point diagnostic-oracle headroom. We added this method to the Experimental Setup, Results, Discussion, Conclusion, and Appendices B and E.
>
> This result changes the interpretation in an important way: the discrepancy is not entirely irreducible, but most of the oracle headroom remains difficult to exploit safely. The rescue/break framing also helps explain why the gain is small: useful corrections require very low break rates.
>
> **Single observed move and interpretation of Top5**
>
> We agree that, because each position contains only one observed human move, Top5 inclusion should not be interpreted as recovering the full set of reasonable human moves. We revised the manuscript to make this limitation explicit in the Abstract, Problem Formulation, Results, Discussion, and Conclusion.
>
> In particular, we now consistently describe Top5 as ranking the observed move among the model’s highest-ranked candidates, not as identifying the complete set of human-plausible alternatives. We also clarify that a Top1 mismatch does not necessarily imply a correctable model-ranking error. It may reflect a correctable error, but it may also reflect unobserved player/session context or natural variation in human choice.
>
> **Broader impact**
>
> We added a Broader Impact section to address the reviewer’s concerns. We note that human-move prediction models may be useful for chess education, training tools, and analysis systems, but player-adaptive systems can involve sensitive behavioral data such as move choices, timing patterns, ratings, and historical play. We therefore emphasize privacy protections, transparency, and data minimization.
>
> We also clarify that move-prediction models should not be treated as standalone evidence in cheating, plagiarism, or other misconduct investigations. Their outputs may provide contextual signals for analysis, but high-stakes judgments should require independent evidence, careful human review, and an opportunity for affected users to contest the decision.

---

### Review · Reviewer_aWWT · 2026-07-10

**Summary Of Contributions:**

**Summary**:

The paper is broadly focused on models for human move prediction in chess. It points out a shortcoming in the top-1 evaluation metric used to assess and compare the models in this area, by pointing to a gap in two evaluation metrics (top-1 vs. top-k/top-5). The paper defines a discrepancy metric between the top-1 and top-k metrics, and refers to it as the “argmax gap”. It also defines a couple of additional metrics (Rescue and Break) to compare two models, and decomposes the difference in two models' top-1 predictions into the differences in their Break and Rescue decisions, and later uses them to argue that the argmax gap is not trivial to close using existing models' complementary abilities/predictions. The paper empirically characterizes these metrics for two models (MAIA3 and Allie policy-only) and also for some ideas of using the two models in a complementary way, to argue that the argmax gap is persistent, that model performance differs across move times, and to advocate for using a mixture of metrics collectively for model comparison.



**Strengths**:
- The paper empirically assesses how improving NLL does not translate into improving top-1 accuracy. It establishes that we should not expect the two goals to align in human chess move prediction.

**Weaknesses**:
- It does seem that the ongoing research in this area *is* advocating for a shift from the deterministic top-1 metric, for various reasons (behavioral characteristics, cognitive noise, optimizing vs. satisficing, etc.). And, existing works often report factors other than move prediction accuracy/top-1 in their performance evaluations. Therefore, while the persistent discrepancy between top-1 and top-5/top-k metrics as characterized here is of interest, the research community on this topic does seem to be diversifying its evaluation metrics, and not necessarily relying on one metric as its diagnostic tool. With this in view, the existing metrics for evaluation used in the literature, and the value of the empirical characterization of this argmax gap and its persistence needs to be more carefully established.
- Perhaps more fundamentally, while it is the case that move prediction accuracy is not aligned with other evaluation metrics (like top-k), it is unclear *why* we should care about other metrics. What is the purpose of human chess move prediction, if not move prediction accuracy (at most personalized to different users)? It is unclear if this paper is establishing that there is an upperbound on this metric, and that is why we should switch to other metrics as *secondary* comparison metrics; OR, whether it is arguing that there is fundamentally a better metric, or collection of metrics, which should really be the purpose of human chess move prediction.

**Audience:**

Yes

**Audience Explanation:**

Yes, machine learning research on human chess move prediction is an active research area and would be of interest to the TMLR audience.

**Broader Impact Concerns:**

No broader impact concerns.

**Claims And Evidence:**

Yes

**Claims Explanation:**

Yes, to the best of my understanding, the claims of this paper are well supported through a range of empirical results, presented both in the main paper and in the appendices. Code and data will be made available if the paper is accepted.

**Requested Changes:**

1. Can you please clarify what evaluation metrics the existing literature has been using? It is very true that top-1 is the primary metric, but there are a number of works that have used (and proposed) alternative metrics, and these are not identified/discussed here.
2. Are the results of this paper meant to argue that there is an upper bound on how much the top-1 prediction can possibly improve? Are they arguing that top-1 is a bad metric for this area anyway, and fundamentally, something other than top-1 should be the real goal of human chess move prediction? Otherwise, yes, there can indeed be a gap between top-1 and top-k predictions that is non-trivial to cover, but perhaps the real goal should still be top-1 prediction accuracy, and whether it has a gap with other metrics is true yet those other metrics should not or will not become a target in any case.
3. The reason for the choice of MAIA3 and Allie for conducting the comparisons could benefit from more explicit support.
4. Is it correct to say that the argmax gap in this paper is used as a diagnostic tool? If yes, is there a way to use it as an optimization tool? More broadly, can the authors comment on whether they recommend a different training objective based on these findings, if that is even feasible? If my understanding is correct that the argument is that NLL optimization/refinement does not improve top-1 accuracy (loss and accuracy are not always aligned, especially when the outcome is probabilistic, which seems expected), is there something better to do for human chess prediction?
5. Small typo: Missing period page 2, the very end of the paragraph starting with “The two models also contain complementary information…”

---

> ### Author Response · Authors · 2026-07-16
> **Response to Reviewer aWWT**
>
> We thank the reviewer for the thoughtful assessment. We have revised the manuscript to better explain how our analysis relates to existing evaluation practice, why MAIA3 and Allie are the focus of the comparison, and how the argmax-gap diagnostic can guide improvement attempts.
>
> **Existing evaluation metrics and the role of Top1**
>
> We agree that the human-chess-modeling literature does not rely only on Top1 accuracy. We revised the Introduction to make this clearer. In particular, we now note that exact move matching remains central, but related work also studies skill-conditioned behavior, move-prediction perplexity and coherence, game-log behavior such as thinking time, and player-specific adaptation. This revised framing is meant to position our contribution more precisely: we are not introducing Top-k accuracy as a new idea, and we are not arguing that the field has ignored all metrics other than Top1. Instead, we study how rank-based accuracy, probability quality, and final move selection relate in strong current predictors.
>
> We also clarified the role of Top1. We are not arguing that Top1 is a bad metric or that it should be replaced. When the goal is to predict the move that was actually played, Top1 remains the most direct metric. Our point is that Top1 alone does not explain why a prediction failed: it does not show whether the observed move was far down the ranking or already near the top, whether the model assigned it a reasonable probability, or whether an intervention helped more positions than it hurt. We therefore revised the Discussion and Conclusion to frame Top-k, NLL, calibration, MRR/NDCG, and rescue/break behavior as diagnostic complements to Top1 rather than replacements for it.
>
> **No upper-bound claim**
>
> We also clarified that the argmax gap should not be read as an upper bound on achievable Top1 accuracy. The revised formulation describes the argmax gap as a descriptive summary of the empirical Top5–Top1 discrepancy. In the Conclusion, we now state that stronger models, richer player/session context, and different training or reranking objectives may further improve exact move matching. The experiments show that the gap is difficult to close with the tested methods, not that it is impossible to close.
>
> **Choice of MAIA3 and Allie**
>
> We expanded the motivation for choosing MAIA3 and Allie. They are a natural pair for this study because they are strong public human-move policies from distinct modeling lines, both produce legal-move distributions needed for rank-based and probability-based evaluation, and both can be evaluated under the same shared Allie blitz protocol. This lets us compare their predictions, likelihoods, Top-k ranks, and rescue/break behavior on exactly the same positions and legal move sets.
>
> **Argmax gap as diagnostic and as a guide for improvement**
>
> We agree with the reviewer that it is important to clarify whether the argmax gap is only descriptive or whether it can guide better methods. We now frame it primarily as a diagnostic tool, but we also added a direct improvement attempt based on that diagnostic. The rescue/break analysis suggests that useful interventions must be conservative: they need to correct base-model errors without changing too many predictions that are already correct. Motivated by this, we added a conservative rank-2 correction gate that keeps MAIA3’s Top1 prediction by default and switches only to MAIA3’s second-ranked move under a validation-selected pre-move rule.
> | Method                   |   Top1 | Delta vs. MAIA3 | Rescues | Breaks |    Net | Switch rate |
> | ------------------------ | -----: | --------------: | ------: | -----: | -----: | ----------: |
> | Conservative rank-2 gate | 57.391 |       +0.137 pp |  14,176 | 12,969 | +1,207 |      5.338% |
>
> The gain is small but reliable: both the position-level and 10k game-clustered 95% intervals are [+0.100, +0.173] percentage points. This recovers 3.01% of the 4.541 percentage-point diagnostic-oracle headroom. We added the method to the Experimental Setup, Results, Discussion, Conclusion, and Appendix B and E. The result supports a more nuanced conclusion: the discrepancy is not entirely irreducible, but most of the oracle headroom remains difficult to exploit safely.
>
> More broadly, we now discuss that future methods may need objectives or post-processing rules that combine distributional quality with decision-aware constraints on when to change a strong base model’s Top1 prediction. We do not claim to provide a complete replacement training objective, but the rank-2 gate shows that the diagnostic can guide a small, measurable improvement.
>
> **Typo**
>
> In the revised version, we fixed the missing period noted by the reviewer.

---

### Review · Reviewer_DFXt · 2026-07-10

**Summary Of Contributions:**

As stated in the paper, there are 4 contributions
1. The paper empirically characterizes the "argmax gap", which is the discrepancy between a model ranking a human move in the Top5 versus choosing it as the Top1 prediction. It also  introduces a rescue/break accounting framework to track how net accuracy changes.
2. The authors quantify the complementary predictive signals between the MAIA3 and Allie models. They demonstrate that various pre-move selectors and ensembles recover very little oracle headroom because newly introduced errors offset corrected ones.
3. They show that prediction performance varies significantly across game contexts.
4. They establish that better probability modeling does not necessarily translate to better exact move matching. Calibration, trust-region refinement, and ensembling improve negative log-likelihood with little to no corresponding gain in Top1 accuracy

**Additional Comments:**

1. Top1 is stated in the intro, without clearly defining it.  May not be familiar to audience not working on the problem.

2.  A toy example to walk through the mathematical formulation would help the reader.

3. The data visualizations, specifically the cumulative rank and the rescue/break partitioning are highly intuitive

**Audience:**

Yes

**Audience Explanation:**

The findings presented in the paper stretch beyond chess. The core finding that optimizing a model's likelihood (NLL) can completely decouple from improving its discrete top-1 decision accuracy is a highly relevant to anyone working on problems of Human-AI alignment, LLM decoding strategies, etc.

**Broader Impact Concerns:**

There are no significant ethical concerns  that would require the addition of a Broader Impact Statement for this submission.

**Claims And Evidence:**

Yes

**Claims Explanation:**

Based on the provided manuscript, yes, the claims made in the submission are supported by accurate, convincing, and clear evidence. The paper operates with exemplary methodological hygiene. The scale of the dataset (884,049 positions), the use of two entirely independent base models (MAIA3 and Allie), strict split-independence checks (zero game or row leakage), and the introduction of the exact mathematical identity of "Rescue/Break accounting"  validate reported findings.

**Requested Changes:**

1. In the Abstract and Introduction, the text broadly states that "pre-move learned selectors... recover little of this headroom". However, Appendix B reveals that the learned selectors are implemented exclusively as linear models. To ensure scientific precision, the authors must qualify these statements in the Abstract, Section 1, and Section 6. They should explicitly specify that the conclusions apply to the tested linear learned selectors.

2. It would vastly strengthen the paper's core thesis if the authors ran a quick baseline using a standard non-linear model—such as a Gradient Boosted Decision Tree (e.g., XGBoost) or a small Multi-Layer Perceptron (MLP)—on their existing 45-feature matrix.

3. The paper cleanly demonstrates that probability modifications optimize NLL but fail to alter Top1 accuracy. However, it remains unclear if these interventions are completely stagnant, or if they are successfully pushing the human move closer to the top of the shortlist. Reporting Mean Reciprocal Rank (MRR) or Normalized Discounted Cumulative Gain (NDCG) across Table 4 (Probability-level modifications) would provide a valuable midpoint analysis between a strict, hard argmax (Top1) and a full distributional fit (NLL).

4.  In Section 5, the authors note the single-observation constraint (each position only gives us one human move). The paper would be strengthened by expanding the qualitative discussion on whether exact Top1 matching is an inherently flawed objective in human modeling.

---

> ### Author Response · Authors · 2026-07-16
> **Response to Reviewer DFXt**
>
> We thank the reviewer for the positive assessment and for their suggestions. We have revised the manuscript to improve the precision of the learned-selector claims, added nonlinear selector baselines, added MRR/NDCG as intermediate ranking metrics, and expanded the discussion of Top1.
>
> **Scope of learned-selector claims and nonlinear baselines**
>
> We agree that the submitted wording around “pre-move learned selectors” was too broad. In the submitted version, the learned selectors were linear/logistic models. We have revised the Abstract, Introduction, Results, Discussion, Conclusion, and Appendix B to specify the tested selector families.
>
> Following the reviewer’s suggestion, we also added nonlinear baselines on the same audited 45-feature pre-move/model-output matrix, excluding realized target-move duration and all derived timing features. All model families, hyperparameters, and thresholds were selected on held-out validation data, with paper-test labels used only after selection.
>
> The nonlinear baselines improve some selector variants relative to the submitted linear versions, but they do not recover substantial oracle headroom over MAIA3 as seen in the table below:
> | Method                           |  Ref. |   Top1 | Delta vs. Ref. | Delta vs. MAIA3 | Rescues | Breaks |    Net |
> | -------------------------------- | ----: | -----: | -------------: | --------------: | ------: | -----: | -----: |
> | Cross-model MLP selector         | MAIA3 | 57.083 |      -0.172 pp |       -0.172 pp |  16,835 | 18,358 | -1,523 |
> | Cross-model XGBoost selector     | MAIA3 | 57.097 |      -0.158 pp |       -0.158 pp |     743 |  2,140 | -1,397 |
> | MAIA3 shortlist XGBoost selector | MAIA3 | 57.090 |      -0.165 pp |       -0.165 pp |      77 |  1,532 | -1,455 |
> | Allie shortlist XGBoost selector | Allie | 56.422 |      +0.688 pp |       -0.833 pp |  10,914 |  4,828 | +6,086 |
>
> Thus, the conclusion is that the tested linear, small-MLP, and XGBoost learned selector baselines do not recover meaningful oracle headroom over MAIA3. We report these results in the main Results section and provide statistical validation in Appendix E.
>
> **MRR/NDCG as intermediate ranking metrics**
>
> We added the requested midpoint ranking metrics for the probability-level methods. We now compute MRR and NDCG@k using the same strict-rank convention as the Top-k metrics. The results support the paper’s distinction between distributional fit, intermediate ranking, and exact Top1 recovery:
> | Comparison                   | Delta NLL | Delta Top1 | Delta MRR | Delta NDCG@5 |
> | ---------------------------- | --------: | ---------: | --------: | -----------: |
> | MAIA3 calibrated vs. base    | -0.008138 |  +0.000 pp | +0.000000 |    +0.000000 |
> | MAIA3 trust-region vs. base  | -0.007176 |  -0.167 pp | -0.001162 |    -0.001068 |
> | Allie calibrated vs. base    | -0.010096 |  +0.000 pp | +0.000000 |    +0.000000 |
> | Allie trust-region vs. base  | -0.008656 |  +0.000 pp | -0.000001 |    -0.000007 |
> | Convex ensemble vs. MAIA3    | -0.003573 |  +0.016 pp | +0.000106 |    +0.000057 |
> | Geometric ensemble vs. MAIA3 | -0.004987 |  -0.008 pp | +0.000006 |    +0.000029 |
>
> Calibration improves NLL but leaves MRR and NDCG exactly unchanged because it is rank-preserving. MAIA3 trust-region refinement improves NLL but slightly worsens Top1, MRR, and NDCG. The convex and geometric ensembles produce tiny positive MRR/NDCG point changes, but their confidence intervals include zero and their Top1 changes remain negligible. We added the MRR/NDCG table to the main Results and the paired deltas, confidence intervals, and rank-change counts to Appendix E.
>
> **Role of Top1**
>
> We are not arguing that Top1 should be discarded. In fact, when the goal is to predict the move that was actually played, Top1 remains the most direct metric. Our point is that Top1 alone does not tell the whole story. It does not show whether the observed move was far down the ranking or already near the top, whether the model assigned it reasonable probability, or whether a change in prediction helped more positions than it hurt. Since each position gives us only one observed human move, a Top1 mismatch can reflect a correctable ranking error, missing player/session context, or natural variation in human choice. The revised Discussion therefore recommends reading Top1 together with Top-k ranking, likelihood, calibration, MRR/NDCG, and rescue/break behavior.
>
> We also thank the reviewer for the positive comment on the cumulative-rank and rescue/break visualizations; we retain these visualizations in the revised manuscript.